# Generative Topological Networks

## Abstract

Generative methods have recently seen significant improvements by generating in a lower-dimensional latent representation of the data. However, many of the generative methods applied in the latent space remain complex and difficult to train. In this work, we introduce a new and simple generative method grounded in topology theory –– *Generative Topological Networks (GTNs)* – that is designed to operate on lower-dimensional latent representations of the data. GTNs are simple to train – they employ a standard supervised learning approach and do not suffer from generative pitfalls such as mode collapse, posterior collapse or the need to pose constraints on the neural network architecture. We demonstrate the use of GTNs on several datasets, including MNIST, CelebA, CIFAR-10 and the Hands and Palm Images dataset by training GTNs on a lower-dimensional latent representation of the data. We show that GTNs can improve upon VAEs and that they are quick to converge, generating realistic samples in early epochs. Further, we provide several avenues for future research. For example, we show how GTNs can be adapted to accommodate for more sophisticated data distributions that include disconnected components. We also discuss potential algorithmic and architectural improvements that are worth investigating. Finally, we note that the topological observations provided herein may also be of value for other methods.

## 1 Introduction

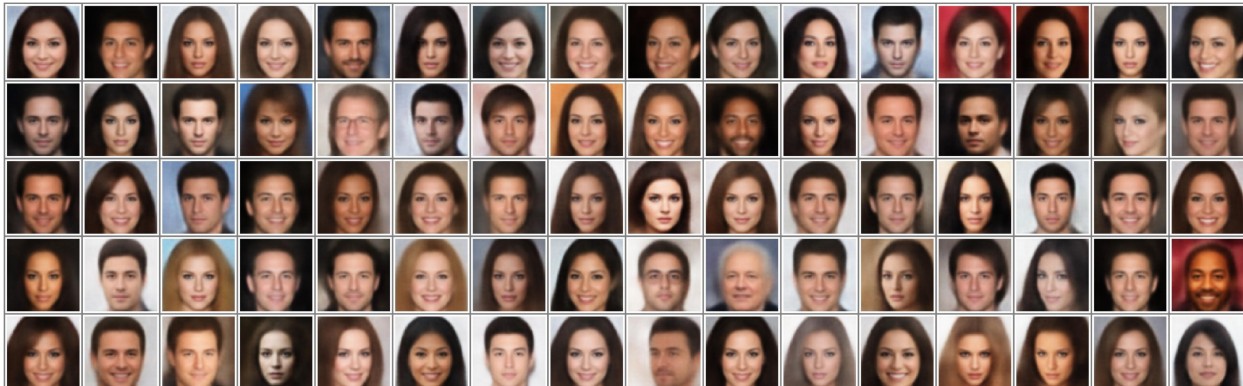

Figure 1: Samples generated by a GTN trained on a latent representation of CelebA $64 \times 64$ with latent dimension $d = 100$.

Deep generative models such as Generative Adversarial Networks (GANs) (Goodfellow et al., 2020), Variational Autoencoders (VAEs) (Kingma & Welling, 2013), Energy-Based Models (EBMs) (LeCun et al., 2006; Ngiam et al., 2011), normalizing flows (NFs) (Rezende & Mohamed, 2015), and diffusion models (Sohl-Dickstein et al., 2015a), have demonstrated remarkable capabilities for generating samples based on training data distributions (Kang et al., 2023; Ho et al., 2020; Ramesh et al., 2021; Ho et al., 2019; Kingma & Dhariwal, 2018; Kingma et al., 2016; Reed et al., 2017; Van Den Oord et al., 2017; Ramesh et al., 2021; Onken et al., 2021).

Many image-generation methods like diffusion are often applied in pixel-space (Ho et al., 2020; Sohl-Dickstein et al., 2015b). More recently, generative methods have seen great improvements in generative quality by transitioning to a lower-dimensional latent representation of the data (Rombach et al., 2022). Although training in a lower-dimensional latent space reduces the computational burden, the methods used to generate in the latent space remain complex and computationally expensive.

In this work we introduce a new class of generative models – Generative Topological Networks (GTNs) – that provides a simple approach for generating in the latent space. Specifically, given a training set of samples (e.g. images) and a tractable source distribution (e.g. Gaussian), GTNs learn to approximate a continuous and invertible function $h$ such that, given a $y$ sampled from the source distribution, $h(y)$ is a sample representing the training data distribution. GTNs are reminiscent of NFs, which aim to transform one distribution into another using a sequence of invertible (and differentiable) maps. NFs, however, pose specific constraints on the network architecture. In contrast, GTNs do not pose any architectural constraints, and are fully operational using a simple vanilla fully-connected architecture.

From a **practical** perspective, GTNs are simple to train, requiring only a single, vanilla, feedforward architecture trained using standard supervised learning. This allows GTNs to: avoid issues like mode collapse or posterior collapse faced by GANs and VAEs; circumvent the intricacies of training more complex architectures such as those employed by diffusion and GANs; and avoid posing constraints on the structure of the neural network, as in NFs. These advantages manifest in our experiments – with realistic samples obtained at early epochs using a single T4 GPU.

From a **theoretical** perspective, GTNs provide guarantees and properties that are desirable in the context of generative models. These include: learnability (via the universal approximation theorem), continuity (for continuous interpolations – Figure 6), bijectivity (for diversity and coverage of the data distribution – Figure 3) and topological properties that serve as guiding principles that can inform the design of generative methods (see the Method section and the swiss-roll example). We also discuss the importance of these topological properties in the context of other methods too.

The remainder of the paper is structured as follows: In the Method section we first develop the theory behind GTNs for the 1-dimensional (1D) case and use it to accurately generate samples from an intrinsically 1D swiss-roll represented in a 2D ambient space. Here, we emphasize the significance of the intrinsic dimension of the data for both GTNs as well as for other generative models like NFs. We then extend the theory behind GTNs from the 1D case to higher dimensions and use it to accurately generate samples from the multivariate uniform distribution. Finally, we apply GTNs to real datasets represented in a lower-dimensional latent space obtained by autoencoders. We show that GTNs generate samples resembling reconstructed data and that they improve upon VAEs both quantitatively and qualitatively. We conclude with Related Work and Discussion sections where we elaborate on how GTNs compare to other methods from a practical perspective and offer several directions for future research that could provide further improvements to GTNs, including a demonstration of an extensions of GTNs to more complicated data.

## 2 Method

We first develop the method in the simple case of a 1D space, and proceed to generalize to higher dimensions.

### 2.1 1-Dimension

Consider a continuous random variable $X$ with values in $\mathbb{R}$. We wish to generate samples from $X$ without knowing its distribution. One solution would be to sample from a known and tractable distribution, such as the standard normal distribution, and then to apply a function that maps this sample to a corresponding sample from $X$. Diffusion models attempt to approximate such a mapping through gradual stochastic manipulations of the standard normal sample back to a sample from $X$. We will show that, under certain general conditions, such a mapping can be explicitly defined, providing a simple deterministic function which we will denote as $h$ (and which we illustrate in Figure 2). We will show that $h$ is in fact a homeomorphism – it is continuous, invertible and has a continuous inverse (see Definition 2.1.1). This has significant implications, as we will soon explain.

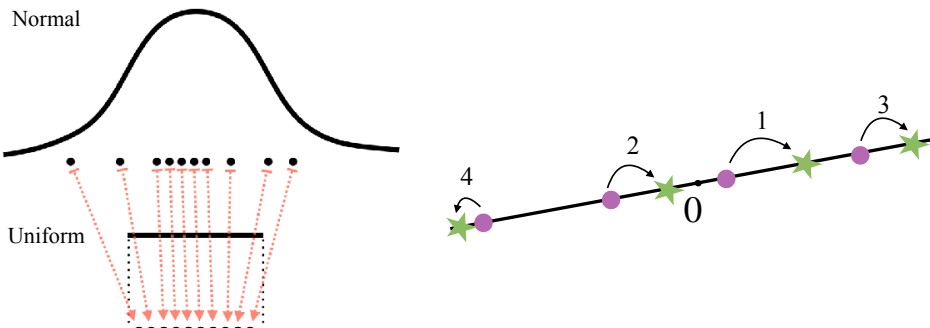

Figure 2: Illustration of the mapping produced by $h$ and of the labeling process for training its approximation $\hat{h}$ (formally described in Algorithm 1). In the 1D illustration (left), $Y$ is normally distributed and $X$ is uniformly distributed. A point $y$ from the normal sample is labeled with the unique point $x_y$ from the uniform sample that has the same empirical CDF value as $y$. In the 2D illustration (right), this occurs on the line passing through the origin (imagine the round points being normal samples and the stars being uniform samples). The numbers reflect the order in which the algorithm matches $y$ (circles) with $x_y$ (stars).

### 2.1.1 Defining $h$.

In this section we will define the aforementioned function $h$ that transforms one distribution into another. We will also prove that $h$, under fairly general conditions, possesses certain properties that are desirable in the context of generative models by proving that it is a homeomorphism (Theorem 2.1.1). For example, we will see that $h$ is bijective, so that each sample $y$ is mapped precisely to one sample $x_y = h(y)$, generating different samples for different $y$, and guaranteeing that each sample $x$ has a sample $y$ that generates it. We will also discuss other useful consequences of $h$ being a homeomorphism. We begin by defining the term *homeomorphism* in our context and proceed to defining $h$ in Theorem 2.1.1.

**Definition 2.1.1** *(Homeomorphism for $\mathbb{R}^n$).* Let $S, T$ be two subsets of $\mathbb{R}^n$. A function $h : S \to T$ is a **homeomorphism** if: (1) $h$ is continuous; (2) $h$ is bijective; (3) $h^{-1}$ is continuous. When such an $h$ exists, then $S$ and $T$ are called **homeomorphic**.

**Theorem 2.1.1** Let $X$ and $Y$ be random variables that have continuous probability density functions (pdfs) $f_X$, $f_Y$ and supports $S_X$, $S_Y$ that are open intervals in $\mathbb{R}$. Denote the corresponding cumulative distribution functions (CDFs) as $F_X$ and $F_Y$. Define:

$$
\begin{aligned}
&h : S_Y \to S_X \\
&h(y) = F_X|_{S_X}^{-1}(F_Y|_{S_Y}(y))
\end{aligned}
\tag{1}
$$

Then: (1) $h$ is well-defined; (2) $h$ is a homeomorphism.

Note that the requirement that $S_X$, $S_Y$ are open intervals can be adapted to a union of open intervals – see Appendix D. The proof of Theorem 2.1.1 is in Appendix A.

The simple special case of $f_X, f_Y > 0$ in $\mathbb{R}$ illustrates the key ideas of Theorem 2.1.1. In this case we have that:

$$
\begin{aligned}
&h : \mathbb{R} \to \mathbb{R} \\
&h(y) = F_X^{-1}(F_Y(y))
\end{aligned}
\tag{2}
$$

For simplicity of exposition, we continue with this special case but note that adapting the results to the general case is a matter of technicality. Namely – $f_X, f_Y > 0$ on $S_X, S_Y$ respectively. This, combined with

$S_X$ and $S_Y$ being open intervals, means that the restricted CDFs $F_X|_{S_X}, F_Y|_{S_Y}$ are continuous and strictly monotonically increasing on $S_X, S_Y$, which is the main observation needed in order to generalize.

On the many advantages of $h$ being a homeomorphism in the context of generative models see Appendix I.

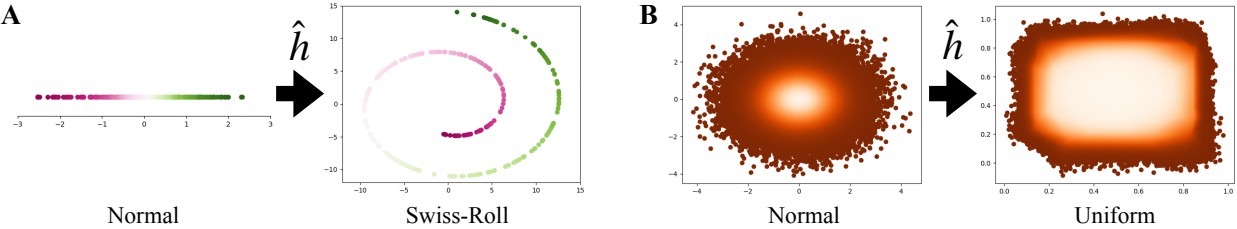

Figure 3: **(A)** Test results for a GTN $\hat{h}$ trained to map from $Y \sim \mathcal{N}(0, 1)$ to the swiss-roll parameter. The color indicates which point in the normal sample was mapped to which point in the swiss-roll. **(B)** Test results for a GTN $\hat{h}$ trained to map from $Y \sim \mathcal{N}(\mathbf{0}, \mathbf{I})$ to $X \sim U((0, 1) \times (0, 1))$. The color is based on the normal sample (left): for each $y$ in the normal sample, $\hat{h}(y)$ has the same color as $y$ so that the figure on the right shows how the normal sample was stretched to a uniform distribution.

### 2.1.2 Learning $h$ Using a Neural Network $\hat{h}$ and Generating Samples with $\hat{h}$.

As explained in the previous subsection, one consequence of $h$ being a homeomorphism is that it can be approximated by a feedforward neural network $\hat{h}$. If we had known both $F_X^{-1}$ and $F_Y$ then we would be able to easily generate labels for each $y \in Y$: denoting the label of a given $y \in Y$ as $x_y$, we would set $x_y = h(y)$, meaning:

$$x_y = h(y) = F_X^{-1}(F_Y(y)) \tag{3}$$

Although we do not know $F_X$, $F_Y$, we do have access to samples from $X$ and $Y$ which we can use to approximate $x_y$. From Eq. 3 we see that $x_y$ satisfies $F_X(x_y) = F_Y(y)$, meaning $x_y$ is the unique $x \in X$ that matches the percentile of $y$. We can use this observation to approximate $x_y$ by replacing $F_X$ and $F_Y$ with the empirical CDFs as follows.

Let $D_X := \{x_1, \ldots, x_n\}$ and $D_Y := \{y_1, \ldots y_n\}$ be $n$ observed values of $X$ and $Y$, respectively. Labels are obtained easily by sorting $D_X$ and $D_Y$ to obtain $D_X^{\text{sorted}}$ and $D_Y^{\text{sorted}}$ and labeling each $y \in D_Y^{\text{sorted}}$ with the corresponding $x \in D_X^{\text{sorted}}$ (i.e. assuming $D_X, D_Y$ are sorted, $y_i$ is assigned $x_i$). This is illustrated in Figure 2.

The loss function is the MSE:

$$\frac{1}{n} \sum_{i=1}^{n} ||\hat{h}(y_i) - x_{y_i}||^2$$

Generating samples from $X$ is now straightforward – we sample $y \in Y$ and compute $\hat{h}(y)$.

### 2.1.3 Example: Swiss-Roll.

In this example, we show that samples from the swiss-roll can be easily generated from a 1D normal distribution with visibly near-perfect accuracy (Figure 3 (A)) using a GTN. We first explain why this would not be as easy using a 2D normal distribution.

The swiss-roll is a 1D manifold in a 2D space $\mathbb{R}^2$ (generated using a single, 1D, continuous random variable $\theta$). Being a 1D manifold, it cannot be homeomorphic to the support of a 2D normal distribution ($\mathbb{R}^2$) since homeomorphic manifolds must have the same dimension. Therefore, there would be no hope of learning a homeomorphism that maps a 2D Gaussian random variable to the swiss-roll. However, the swiss roll *is*

homeomorphic to $\mathbb{R}$ – the support of a 1D standard normal distribution. In fact, since $\theta$ is a random variable that satisfies the conditions in Theorem 2.1.1, it is homeomorphic to $\mathbb{R}$ via $h$, which we can learn using $\hat{h}$. The fact that there is no hope of learning a homeomorphism in a higher-dimensional space than the intrinsic dimension is also important for other generative methods like NFs, which use a diffeomorphism (a type of homeomorphism), and it may also help explain why latent diffusion models have shown improvements over pixel-space diffusion models (Rombach et al., 2022).

After justifying the use of a 1D Gaussian, we return to our example. To train $\hat{h}$, we created a dataset of $n = 50,000$ samples from $\theta$ (Appendix H), denoted $D_X$. We sampled $n$ samples from a 1D standard normal distribution to obtain $D_Y$. We labeled each $y_i \in D_Y$ with its $x_{y_i} \in D_X$ as defined in the previous section. We used a standard feedforward neural network as $\hat{h}$ (4 layers of width 6; details in Appendix Table 5). Figure 3 (A) shows the result of testing the trained model $\hat{h}$ on a set of new samples $y_1, \ldots, y_k$ drawn from $\mathcal{N}(0, 1)$ (each point is obtained by predicting $\hat{\theta}_i := \hat{h}(y_i)$ and applying the formula for the swiss roll to $\hat{\theta}_i$ (Appendix H).

Besides demonstrating that $\hat{h}$ can serve as an accurate generative model, this example also emphasizes an important point – namely, that if we want to learn such a homeomorphism $h$, we might need to reduce the dimensionality of the data if it is not already in its intrinsic dimension. This observation will guide us in later sections.

## 2.2 Higher Dimensions

To generalize to more than one dimension, we reduce to the 1D case by considering lines that pass through the origin. Briefly, we take the random variables obtained by restricting $X$ and $Y$ to each line, and apply the homeomorphism defined for the 1D case to these random variables. We begin with a formal setup that is very similar to the 1D case.

### 2.2.1 Defining $h$.

Let $X = (X_1, \ldots, X_d), Y = (Y_1, \ldots, Y_d)$ be multivariate random-variables with continuous joint probability density functions $f_X, f_Y$, and with supports $S_X, S_Y$, each of which is a product of $d$ open intervals in $\mathbb{R}$. For example, $X$ could be uniformly distributed with support $S_X = (0, 1) \times (0, 1)$ and $Y$ could be normally distributed with support $S_Y = \mathbb{R} \times \mathbb{R}$.

Consider first rotation invariant distributions (e.g., the standard multivariate normal). That is, assume that for every rotation matrix $R$ and every $x \in \mathbb{R}^d$, $f_X(Rx) = f_X(x)$, and similarly for $f_Y$. For simplicity, also assume that $S_Y, S_X = \mathbb{R}^d$. We can now define the following homeomorphism:

$$
\begin{aligned}
& h : \mathbb{R}^d \to \mathbb{R}^d \\
& h(y) = \begin{cases} h_1(||y||) \frac{y}{||y||}, & y \neq 0 \\ 0, & y = 0 \end{cases}
\end{aligned}
\tag{4}
$$

where $h_1 : (0, \infty) \to (0, \infty)$ is the 1D homeomorphism applied to the random variables $||Y||, ||X||$.

We prove that this is indeed a homeomorphism in Appendix E. Note that this generalizes the 1D case since for $d = 1$ we get $y \mapsto h_1(y)$. Intuitively, $h$ can be seen as shrinking or stretching $y$ to the unit vector in $y$'s direction ($y/||y||$), and then shrinking or stretching it to reach the $x_y := h(y)$ that has the same ranked distance as $y$ on the line (by multiplying it by $h_1(||y||)$). More precisely, $h_1$ produces $x_y$'s distance from the origin so that it has the same quantile as $y$'s distance from the origin (when measured with respect to the random variables obtained by restricting $Y, X$ to the line segment from the origin in $y$'s direction). Another way of thinking about this is provided in Appendix F.

Note that more complicated distributions can be used, as demonstrated in Figure 7 and in the 2D uniform case, which is non-rotation-invariant (Figure 3). This is also discussed in Appendix D. However, for such

---

**Algorithm 1** Labeling

**Input**:
$D_X = \{x_1, \ldots, x_n\}$
$D_Y = \{y_1, \ldots, y_n\}$ sampled from $\mathcal{N}(\mathbf{0}, \mathbf{I})$
**Output**: *res*

1: $res \leftarrow []$
2: $D_X^{\text{sorted}}, D_Y^{\text{sorted}} \leftarrow$ sort $D_X, D_Y$ ascending by $|| \cdot ||_2$
3: **while** $D_Y^{\text{sorted}} \neq \emptyset$:
4:      $y \leftarrow D_Y^{\text{sorted}}[0]$
5:      $x_y \leftarrow \arg\max_{x \in D_X^{\text{sorted}}} cosine\_sim(x, y)$
6:      $D_Y^{\text{sorted}} \leftarrow D_Y^{\text{sorted}}[1 \ldots]$
7:      $D_X^{\text{sorted}} \leftarrow D_X^{\text{sorted}} \setminus \{x_y\}$
8:      $res.append\big((y, x_y)\big)$
9: **return** *res*

---

**Algorithm 2** Sampling

**Input**: $\hat{h}$ trained on *res* (see Alg. 1)

1: $y \leftarrow$ sample from $\mathcal{N}(\mathbf{0}, \mathbf{I})$
2: **return** $\hat{h}(y)$

---

distributions, it may be difficult to explicitly define a homeomorphism $h$ since $h_1$ would depend on the line. Nevertheless, the simpler case above provides guidance on how to train a neural network to match between two distributions on each line. Specifically, it gives rise to an algorithm that aims to perform such 'per line' matching, which we will now introduce.

### 2.2.2 Learning $h$.

Let $D_X$ and $D_Y$ be observed values from $X$ and $Y$. Imagine first the infinite data scenario where each line through the origin has points from both $X$ and $Y$ in both directions (see Figure 2, right). We would like to use the 1D labeling scheme on each line that passes through the origin by using $h_1$ in both directions on the line. Practically, this means that we need to identify those points from $D_X$ and $D_Y$ that reside on the same line and in the same direction, and match them based on their empirical quantiles there (note that the effect of this is reminiscent of Optimal Transport (OT) Santambrogio (2015); Lipman et al. (2022)). The former can be accomplished using maximum cosine-similarity, and the latter can be accomplished using distances from the origin. The process is formally described in Algorithm 1 and illustrated in Figure 2. In Appendix C we prove that Algorithm 1 produces a labeling that is consistent with $h$ for any $d \in \mathbb{N}$ if given a sufficiently large random sample or if given a specific ray-sampling procedure for $Y$. Appendix B gives a less formal intuition behind this proof.

To intuitively see why Algorithm 1 approximates $h$, consider first a line with an infinite sample from $X$ and $Y$ in both directions from the origin (see Figure 2, right). The maximum cosine-similarity over all $x \in D_X$ and a $y$ on this line is 1, so the algorithm returns an $x_y$ that is on this line and in $y$'s direction. Because $D_X, D_Y$ are sorted by distance from the origin, ties in cosine-similarity are broken by distance from the origin, so that the first $y$ is matched with the first $x$, the second $y$ with the second $x$ etc. Note that in practice, unless $d = 1$, the probability that any given line contains points from either $D_X$ or $D_Y$ is negligible. Despite this fact, both sorting by norm and using maximum cosine-similarity are required to prevent labeling discontinuities, as we explain in Section 2.2.4. The use of maximum cosine-similarity allows to approximate this desired labeling by using the nearest point available, providing better approximations as the dataset increases.

Using the labeled samples from Algorithm 1, we train $\hat{h}$ – a feedforward neural network – using MSE as the loss function. We then use $\hat{h}$ to generate new $X$ instances just as in the 1D case, formally described in Algorithm 2.

### 2.2.3 Example: 2-Dimensional Uniform Distribution.

Figure 3 (B) shows a sample generated using Algorithm 2 after applying the method to the multivariate uniform distribution $X \sim U(0,1) \times U(0,1)$. Specifically, we created $D_X$ by sampling $n = 100,000$ points from $X$, and created $D_Y$ by sampling the same number of points from $\mathcal{N}(\mathbf{0}, \mathbf{I})$. We then applied Algorithm 1 to $D_X, D_Y$ to generate labeled data. We trained a standard feedforward neural network $\hat{h}$ (6 layers, width 6; details in Appendix Table 5) and used it to generate new samples according to Algorithm 2. The model was trained until convergence on a separately generated validation set with $n = 10,000$. The colors in both images in Figure 3 (B) are based on the distance of the points from the origin in the Gaussian sample (the left image in (B)), so that the image on the right reflects where each point in the Gaussian sample was predicted to 'move' to by $\hat{h}$.

### 2.2.4 Why both sorting $D_X, D_Y$ by norm and taking maximum cosine-similarity are required.

We first explain why labeling using only cosine-similarity-based matching is not sufficient. Besides ensuring that Algorithm 1 is consistent with $h$ in case there are multiple $X$ and $Y$ samples on a given ray (e.g. consider $d = 1$, where sorting both $D_Y$ and $D_X$ guarantees that each point in $D_Y$ is labeled with a point in $D_X$ of the same empirical-quantile), sorting by norm also serves to prevent other discontinuities. These discontinuities may arise even if each ray contains a single point from either $D_X$ or $D_Y$. Figure 4 shows that if $D_Y$ is not sorted, discontinuities in the labeling arise (identified by the crossing arrows) even when each ray contains a single sample. As shown, these discontinues are resolved when sorting is applied. Appendix Figure 8 shows the poor quality of generated samples when training is performed on the dataset obtained by the labeling algorithm without the sorting.

The question then arises as to whether it is possible to label using *only* the norm (i.e. skipping the use of cosine-similarity in Algorithm 1 and matching only by order after sorting by norm). While this could work for same-signed samples in $d = 1$ (either all positive or all negative), if signs are mixed, or if $d > 1$, this gives rise to many 'crossover' discontinuities. For example, consider shrinking the leftmost target point in example A in Figure 4 to be closer to the $x$-axis. Since the bottom source point has the smallest norm, it will be assigned to this target point, resulting in a crossover similar to that shown in A2.

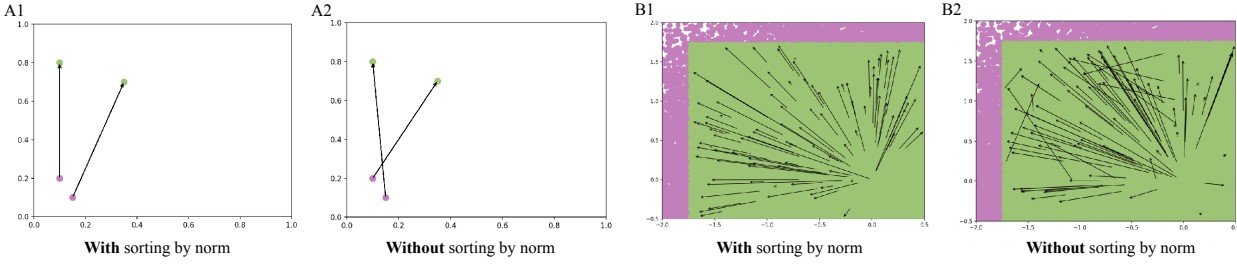

Figure 4: The labeling produced by Algorithm 1 with and without sorting by norm. Each arrow points from a point in the source distribution (purple, also beneath green in B1, B2) to its label point in the target distribution (green). In A1, A2, $D_Y = \{(0.1, 0.2), (0.15, 0.1)\}$, so the second point (bottom purple) has the smaller norm. A1: the resulting labeling with norm-based sorting. The bottom point was labeled (using maximum cosine-similarity) first since the two points were swapped in order when sorted by norm. A2: the resulting labeling without sorting. In B1, B2: Actual labeling obtained for the 2D-uniform distribution example with and without sorting by norm, respectively, for 500 random samples from the upper left quadrant pointing in the outward direction in each. Sorting eliminates the 'crossover' discontinuities observed in B2 (where no sorting was applied).

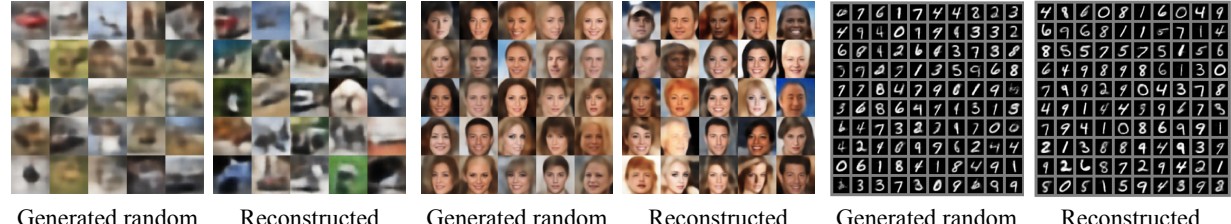

| Generated random | Reconstructed | Generated random | Reconstructed | Generated random | Reconstructed |

Figure 5: Random generated and reconstructed images for CIFAR-10 (left), CelebA (centre) and MNIST (right). Each of the generated samples is the decoded $\hat{h}(r)$ for a random $r \sim \mathcal{N}(\mathbf{0}, \mathbf{I})$) and each of the reconstructions is the decoded vector of a random real image.

## 3 Experiments

In the previous section, we demonstrated how to learn $\hat{h}$ on synthetic data – the swiss-roll and the multivariate uniform distribution. In this section, we will demonstrate how to apply $\hat{h}$ to images.

### 3.0.1 Setup.

In light of the discussion in Section 2.1.3 on using the intrinsic dimension (ID) of the data, we use autoencoders to represent each dataset in a lower dimensional latent space before training the GTN $\hat{h}$. Each dataset was trained separately (with its own autoencoder and $\hat{h}$). Unless mentioned otherwise, all experiments used the same vanilla autoencoder architecture adapted to different latent dimensions (the encoder consisted of two convolution layers with ReLU activation followed by a fully connected layer). For $\hat{h}$ we use a standard feedforward neural network with the width and depth depending on the data. Architecture and training details can be found in Appendix G and Table 5 with further details in our code.

To train $\hat{h}$, we set $X$ to be the latent vectors of the training set (the encoded images), and set $Y$ to be the standard multivariate normal distribution of the same dimension as $X$ (e.g., for a latent dimension of $d = 100$ in the autoencoder, $Y$ has dimension 100). Images were generated using Algorithm 2 by computing: $autoencoder.decoder(\hat{h}(r))$ where $r \sim \mathcal{N}(\mathbf{0}, \mathbf{I})$ (generation occurs in the latent space).

### 3.0.2 Evaluation.

We applied our method to MNIST, CIFAR-10, CelebA $64 \times 64$ and the Hands and Palm (HaP) datasets.

We first applied our method to MNIST (LeCun et al., 1998) using a latent dimension of $d = 5$ to allow for comparison with VAE for the same latent dimension (see Figure 5(b) in Kingma & Welling (2013)). Figure 5 (right) shows random sets of generated images for MNIST, as well as a sample of random reconstructed images for comparison.

Next we applied the method to CelebA (Liu et al., 2015) (Figure 1 and Appendix Figure 13). Images were center cropped to $148 \times 148$ and resized to $64 \times 64$. We tested latent dimensions of $d \in \{10, 25, 50, 100\}$, which are within the range of the typical ID estimated for image datasets (10-50) or close (100) (Pope et al., 2021). We used Inception Score (IS) (Salimans et al., 2016), a common evaluation metric, as the stopping criteria: after every epoch, the GTN generated 200 random images for which the IS was evaluated. We used IS instead of validation-set results since the IS continued to improve well beyond the point of plateau on the validation set. Training stopped after 300 epochs of no improvement in IS. For evaluation, we used both IS and Fréchet Inception Distance (FID) (Heusel et al., 2017). For IS, in addition to the best IS obtained by the model, we produced the IS for a single random set of 200 reconstructed images ("recon-IS") since this reflects the best possible IS that we can reasonably hope to achieve. Plotting IS and recon-IS by epoch, (Appendix Figure 11) shows that IS increases throughout the training process, and that recon-IS is either achieved ($d \in \{10, 25\}$) or nearly achieved ($d \in \{50, 100\}$) by the GTN. For FID, the lowest FID was for

| Data | Method | IS ↑ | recon-IS ↑ | FID ↓ | recon-FID ↓ |
|------|--------|------|------------|-------|-------------|
| CelebA | VAE | 1.0 | 1.0 | 94.66 | 83.48 |
|        | GTN | 1.90 | 1.94 | 73.18 | 67.08 |
| CIFAR-10 | VAE | 1.77 | 1.90 | 286.49 | 277.48 |
|          | GTN | 2.07 | 2.24 | 238.62 | 181.53 |

Table 1: Controlled comparison of GTN with VAE (Kingma et al., 2016) on CelebA and CIFAR-10. The VAE result was obtained by training the vanilla VAE architecture supplied in AntixK (2024). Prior to training the GTN we trained a new autoencoder that employs the same architectures as the vanilla VAE, with only the necessary adaptations to transition from VAE to a vanilla autoencoder. GTN is trained on the latent representations provided by this autoencoder, with $d = 100$ and $d = 128$ for CelebA and CIFAR-10, respectively. We use 10,000 random training images and an equal number of random generated samples (decoded generated latent vectors) to compute FID. We use 200 random generated and 200 random real images for IS.

$d = 100$ with 66.05 and the highest was for $d = 10$ with 119.46. FID and IS across all dimensions are reported in Appendix Table 2.

Observing the progress in image generation during training (Appendix Figure 13), shows that realistic images were already obtained at half the training time (see epoch 276). One epoch took just under 1 minute (50.6 seconds on average) on a single T4 GPU, reflecting 9 hours until the last improvement in IS at epoch 640 and < 4 hours to reach epoch 276. The relatively fast convergence raises the question whether the latent space was approximately normal to begin with. Appendix Figure 15, which compares images generated by decoding random normal vectors with images generated using the GTN, demonstrates that this is not the case.

Next, we designed a controlled experiment to compare our method to the closely-related VAE on both CelebA and CIFAR-10. We used the vanilla VAE suggested for CelebA from (AntixK, 2024) adapting only the latent dimension, and trained two autoencoders: (1) The suggested VAE and; (2) A vanilla autoencoder later used to train a GTN on the learned latent representations. The transition from VAE to vanilla autoencoder included only the necessary adaptations, so that all remaining architectural considerations and hyper-parameters were identical between the two autoencoders. We then used the latent representations from the vanilla autoencoder to train a GTN, where the GTN had the same architecture as in the earlier experiments. No tuning of any kind was performed, including of the random seed (which was set once at the very start of the entire project). Particularly, the GTN architecture and training settings (for both CelebA and CIFAR-10) were unchanged from the previous CelebA experiments (except for the dimensions of the input, output and latent space). Appendix Figure 16 shows a side-by-side comparison of randomly generated images from both methods. Table 1 provides the FID and IS results, including for reconstruction quality (i.e. recon-FID and recon-IS, which compare reconstructions to real images). A comparison with reconstructions is also shown in Figure 5. Although GTN outperforms VAE on both CelebA and CIFAR-10, the CIFAR-10 results are overall worse compared to the CelebA results. This is likely since: (a) the autoencoder settings were originally suggested for CelebA; and (b) no fine-tuning or architecture search was performed for the GTN when transitioning from the CelebA experiment to CIFAR-10. For additional context with other methods, we also provide Appendix Tables 3 and 4.

We next used the CelebA dataset as well as the Hands and Palm Images (HaP) dataset (Kaggle., 2024) to demonstrate that GTNs indeed provide continuous interpolations. The HaP dataset is a small dataset containing 11,000 images of upward-facing and downward-facing hands. We chose the HaP dataset since generating hands is known to be notoriously difficult for generative models (The New Yorker., 2023; Vox., 2023), let alone interpolating between them. As in CelebA, we trained a GTN for each of the four dimensions, on images resized to $64 \times 64$.

Figure 6 shows that GTN indeed generates continuous interpolations. Each row contains generated interpolations between two images (leftmost and rightmost). Specifically, each row contains the results of decoding

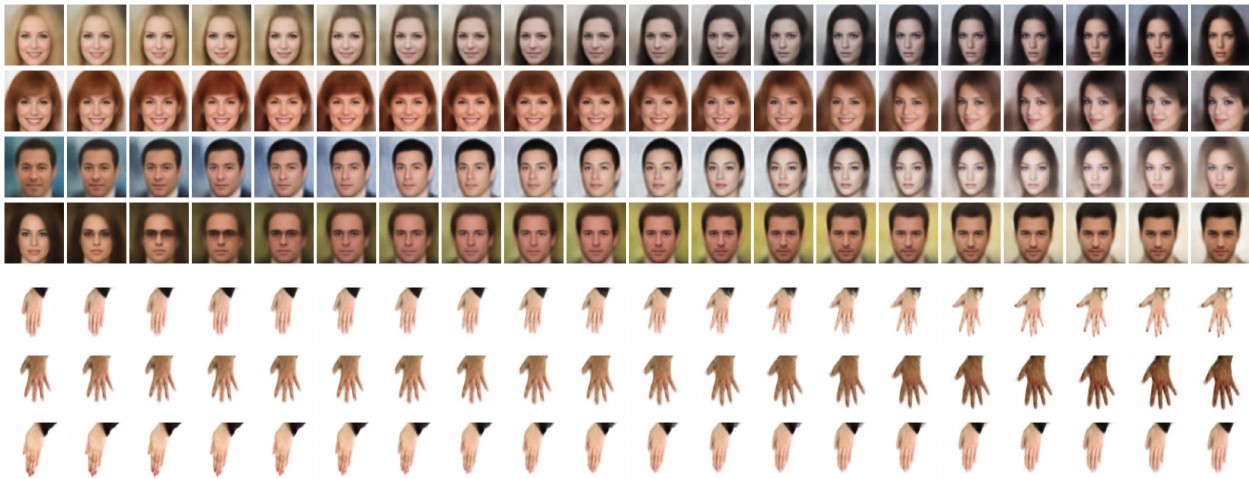

Figure 6: Interpolations generated by a trained GTN for CelebA (top) and the HaP dataset (bottom).

$\hat{h}(\lambda y_{\text{left}} + (1 - \lambda)y_{\text{right}})$ for 20 linearly spaced $\lambda \in [0, 1]$, where, in CelebA, both $y_{\text{left}}$ and $y_{\text{right}}$ were sampled from $\mathcal{N}(\mathbf{0}, \mathbf{I})$, and in HaP $y_{\text{left}}$ and $y_{\text{right}}$ are each the normal label of a real encoded image (from Algorithm 1), chosen so that they have the same orientation. For CelebA we used the $d = 100$ model and for HaP we used $d = 50$ (chosen after consulting the IS and FID results across all four $d$ values in Appendix Table 2 while preferring a higher dimension ($d \in \{50, 100\}$) to retain more visual detail).

It is worth noting an interesting observation arising from the HaP dataset. Notice that the generated interpolations in Figure 6 are between hands of the same orientation (downward facing). However, generated interpolations between hands (downward facing) and palms (upward-facing) do not seem as natural (Appendix Figure 14). This is likely because the dataset does not contain in-between positions for the latter (transitions between downward and upward), but it *does* for the former (closed/open fingers to various degrees). This demonstrates that, regardless of methodology, the completeness of the dataset is important for accurate image generation, and particularly for interpolation.

## 4 Related work

GTNs are related in concept to NFs which seek to map a source distribution into a target distribution using a sequence of bijective transformations. These are typically implemented by a neural network, often in the form of at least tens of neural network blocks (Xu & Campbell, 2024) and sometimes more (Xu et al., 2023). The core of NFs is based on each of these transformations being a diffeomorphism – a specific type of homeomorphism that is more constrained than $h$ as it is defined between *smooth* manifolds (as opposed to any topological spaces, including any manifolds) and requires that the function and its inverse are *differentiable* (as opposed to just continuous for general homeomorphisms). NFs also require that the log-determinant of the Jacobian of these transformations is tractable, posing a constraint on the model architecture. GTNs do not pose any limitations on the model architecture. NFs also differ from GTNs in their optimization method since NFs employ a maximum likelihood objective – typically the Kullback-Leibler (KL) divergence while GTNs are trained using MSE. It was observed that optimizing the KL-divergence may be difficult for more complex normalizing flow structures (Xu et al., 2023). NFs may also suffer from low-quality sample generation (Behrmann et al., 2021), partially because the constraint on the Jacobian, besides limiting the model structure, may also lead to issues such as spurious local minima and numerical instability (Golinski et al., 2019).

Continuous normalizing flow (CNF) are a relatively recent type of NFs that were developed to avoid the main constraints posed by NFs – namely the requirements of invertibility and a tractable Jacobian determinant. To avoid these constraints, CNFs use ordinary differential equations (ODEs) to describe the transformations

in NFs. While CNFs have provided certain improvements over NFs, they are still time-intensive (Huang & Yeh, 2021). Improving both the speed and performance of CNFs is an active and promising field of research. In particular, training CNFs using Flow-Matching (FM) Lipman et al. (2022) – a new paradigm for training CNFs – was shown to yield improved performance, specifically when trained with OT probability-paths (FM-OT). As mentioned in Section 2.2.2, the labeling by Algorithm 1 is conceptually similar to the idea behind OT. Also conceptually similar are rectified flows Liu et al. (2022), which aim to learn straight paths connecting the points drawn from source and target probability distributions in a computationally efficient manner.

Another classs of closely related generative models are VAEs since they are based on autoencoders that aim to transform the lower-dimensional latent space into a tractable distribution. The VAE loss function optimizes two terms, namely both the reconstruction error and the error between the prior and posterior distributions. This often leads to training instabilities and to posterior collapse. Despite these training instabilities, methods based on VAEs have been suggested, including Two-Stage VAE (Dai & Wipf, 2019) which first trains a VAE to identify the lower-dimensional manifold, and then a second VAE to transform the learned latent space into a normal distribution. Other methods, combining VAE and flow, have also been suggested, but have seen limited success at improving upon VAE (Xiao et al., 2019) and add architectural limitations as mentioned earlier. Compared to VAE-based methods, GTNs have several practical advantages. One advantage is that GTNs separate between the autoencoder and the generative process, avoiding the intricacies of balancing between the two terms present in the VAE loss function (an in-depth analysis of the issues arising from this balancing can be found in Dai & Wipf (2019)). Practically, this means that one can focus on finding a high-quality latent representation of the data first, and then learn to generate in that latent space. This may be of particular interest in light of the use of pretrained autoencoders in state-of-the-art generative models (Rombach et al., 2022), and in light of the provided evidence herein that much of the error stems from reconstruction quality. GTNs also show no evidence of mode collapse (a known issue with other methods like GANs). These advantages, and the fact that GTNs employ a standard supervised learning approach, makes them more user-friendly and easier to train than many of the existing methods.

Finally, we note that other works have relied on the manifold hypothesis to guide their design of generative models. One example, already mentioned earlier, is Two-Stage VAE Dai & Wipf (2019). Another is Loaiza-Ganem et al. (2022) which also references the 1D swiss-roll. More can be found in a recent survey aimed at increasing the understanding of generative models in the context of the manifold hypothesis Loaiza-Ganem et al. (2024).

## 5 Discussion

This work introduces a new class of generative models – GTNs – that is explicitly designed for the manifold assumption and offers training stability, speed and simplicity. It also highlights how topological properties like the intrinsic dimension of the data and connectedness (Figure 7) can serve as useful, and even essential, drivers for the design of generative models. The interplay between generative models and topological considerations has also been recently emphasized in Loaiza-Ganem et al. (2024).

As a generative method, GTNs offer a simple and stable approach to image generation. Compared to other generative methods like diffusion, VAE, GANs and NFs, GTNs are computationally simpler and more user-friendly since they employ a basic supervised learning approach that does not suffer from many of the training instabilities and specialized architectural requirements posed by other methods. As an autoencoder-based method, GTNs also allow the flexibility of easily replacing the autoencoder should a better one be designed. This is of value in light of the recent turn to using pre-trained autoencoders for data generation (Rombach et al., 2022). VAEs, for example, would require retraining from scratch, which means risking dealing again with potential instabilities that arise from having to balance between reconstruction and generation.

GTN's potential was demonstrated both quantitatively and qualitatively, particularly with demonstrable improvements over VAE. It is possible that with more sophisticated architectures to replace the fully-connected GTNs (e.g. to convolutional-based GTNs) and more careful fine-tuning, further improvements may be attained (especially for CIFAR-10 where no attempts were made to fine-tune the settings or adapt the architecture when transitioning from CelebA). Upgrading from the vanilla autoencoders may also offer im-

provements, perhaps even more significant ones – observing both the qualitative and quantitative similarity between reconstructions and generated samples (e.g. recon-IS being nearly achieved, the similarity between FID and real-FID as well as the similarities observed in Figure 5) shows that a large portion of the error may be attributed to the quality of the autoencoder.

GTNs can also be extended to data that does not immediately satisfy the assumptions on the support of the distribution (like data that is separable into disconnected components). In such a case, it is possible to exploit the assumption that the data lies on a manifold by labeling points 'locally' using a mixture model before training the GTN. We demonstrate this in Figure 7 where the data distribution is supported on two disjoint sets. In future research it would be interesting to investigate the use of this approach on real datasets, particularly CIFAR-10, which is potentially separable into disconnected components due to the presence of different classes. Note that GTNs were also successfully applied to distributions that are not rotation invariant, particularly the uniform distribution. We leave a more thorough discussion and analysis of this to future work.

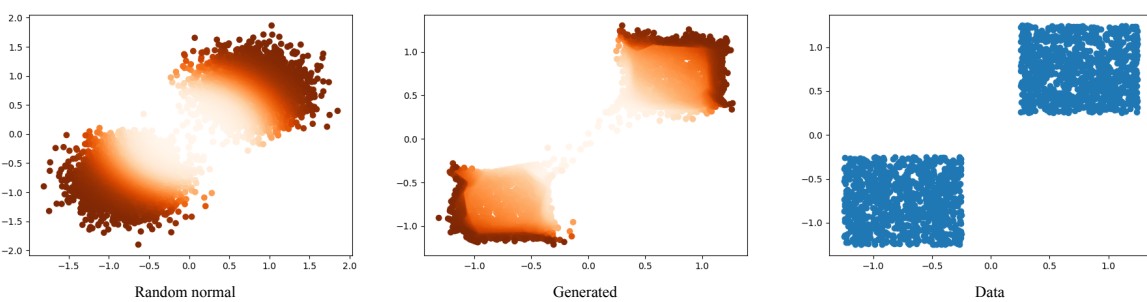

Figure 7: Generating two disjoint uniform distributions. Left-to-right: mixture of random normal samples, corresponding predictions generated by a GTN and actual samples from the data. The random normal samples were generated by clustering the training data into two clusters using the fast-pytorch-kmeans package (DeMoriarty, 2024) and using the cluster means and standard-deviations to define the normal distributions. Labels were computed in each cluster separately according to Algorithm 1 to obtain a single dataset for training the GTN.

Many other avenues for future research exist besides the aforementioned architectural improvements and the extension to more complicated data distributions. One immediate potential improvement may arise from replacing the IS-based stopping-criteria with a different one (IS has been shown to have several shortcomings Barratt & Sharma (2018)). It is also possible that training beyond the point of IS plateau, especially that it is based on a small sample, can produce better generative quality. From a labeling accuracy stand-point, it may also be worth investigating the option of modifying the sampling procedure from $Y$ so as to guarantee the setup used in the proof that Algorithm 1 is consistent with $h$. We expand on this option in the note at the end of Appendix C. Finally, GTNs may also be investigated in the broader context of generative models. For example, it may be used as a component in other generative methods or applied to other data modalities.

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

## Appendix

## A   Proof of Theorem 2.1.1

1. $h$ is well-defined: it suffices to show that: (a) $F_X|_{S_X}$ is invertible, and (b) that the image of $F_Y|_{S_Y}$ is in the domain of $F_X|_{S_X}^{-1}$. For (a), observe that $F_X|_{S_X}$ is continuous (the CDF of a continuous random variable is continuous) , strictly monotonically increasing (the CDF is strictly monotonically increasing on the support) and onto $(0,1)$ (by definition of the CDF and the fact that the support is an open interval). As such it is invertible. A symmetric argument can be made for $F_Y|_{S_Y}$, which we will use later to define the inverse of $h$.

   For (b), we know from the proof of (a) that the image of $F_Y|_{S_Y}$ is the domain of $F_X|_{S_X}^{-1}$.

2. $h$ is a homeomorphism since: a. it is continuous as a composition of continuous functions. Indeed, $F_Y|_{S_Y}$ is continuous (the CDF of a continuous random variable is continuous). $F_X|_{S_X}^{-1}$ is also continuous: since $F_X|_{S_X}$ is continuous (as was $F_Y|_{S_Y}$) and bijective (it is invertible, as shown in the first part) we can use the known result that a continuous and bijective function between two open intervals has a continuous inverse (a consequence of the invariance of domain theorem). ; b. it is bijective since the inverse is $F_Y|_{S_Y}^{-1} \circ F_X|_{S_X}$; c. its inverse is continuous: we know from a. and b. that $h$ is continuous and bijective so we can again use the fact that a continuous and bijective function between two open intervals has a continuous inverse.

## B   Intuition for Algorithm 1

To understand the rationale behind Algorithm 1, imagine four points from each of $X$ and $Y$ lying on the same line, say the $x$-axis in $\mathbb{R}^2$, such that both $X$ and $Y$ have two points on each side of the origin (we assume $D_X, D_Y$ are centralized). See Figure 2 (right), where circles are from $Y$ and stars are from $X$. The cosine similarity of each pair of points $\{a, b\}$ with $a \in D_X, b \in D_Y$ is 1 if they are on the same side or $-1$ of they are on opposite sides. Take the first $y \in D_Y^{\text{sorted}}$ (the closest one to the origin which reflects $\sim$ 50th percentile since we assumed an equal number of points from $Y$ on both sides). The maximum cosine similarity out of all $x \in D_X^{\text{sorted}}$ is 1, meaning $y$ will be matched with an $x$ on the same side. The fact that $D_X$ is sorted, means that we are using the distance as a tie breaker – out of all $x \in D_X$ that are on the same side as $y$, the $x$ that is closest to the origin ($\sim$ 50th percentile) will be chosen as $x_y$. Likewise, the leftmost $y$ will be matched with the leftmost $x$ (both 0th percentile), the rightmost $y$ with the rightmost $x$ (both 100th percentile) etc. Note that if this were in $\mathbb{R}$, this process coincides with the same labeling process described in the 1D case (illustrated in Figure 2). Note that if $D_X, D_Y$ are not sorted, discontinuities in the labeling may arise (see Figures 4 and 8).

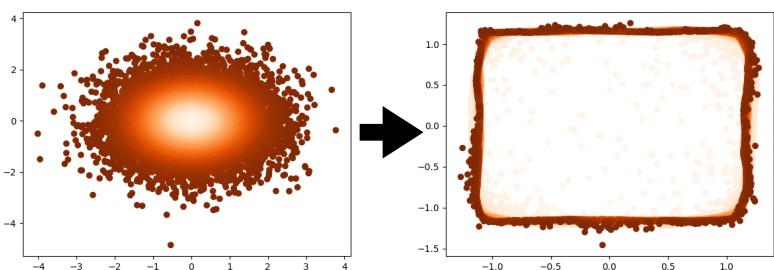

Figure 8: Example demonstrating that sorting $D_X, D_Y$ in the labeling algorithm is essential. Training GTN on the 2D-uniform data on labels produced by the same algorithm with sorting removed causes test predictions to collapse mainly to the boundary.

## C Algorithm 1 is consistent with $h$ under certain assumptions

**Theorem C.1**: Let $X$ and $Y$ be random variables that satisfy the assumptions in Section 4. Assume that $D_X$ and $D_Y$ are such that for every $x \in D_X$ there can be assigned a unique $y \in D_Y$ for which $cos\_sim(x,y) = 1$. That is, if there are $k$ different vectors in $D_X$ on ray $r$, then there are $k$ different vectors in $D_Y$ on ray $r$ (see note below proof on which conditions could allow this to occur). Then Algorithm 1 halts with $res = \{(y, x_y)\}_{y \in D_Y}$ such that for all $y \in D_Y$, $x_y = \tilde{h}(y) = \tilde{h}_1(||y||)\frac{y}{||y||}$ with $\tilde{h}_1$ being the empirical quantile-matching function that approximates $h_1 = F_X^{-1} \circ F_Y$.

**Proof:**

Clearly Algorithm 1 halts after $N = |D_Y^{\text{sorted}}|$ iterations.

Note that two different iterations of Algorithm 1 are independent if they involve different rays. More formally, let $y_i, y_j$ be the vectors from $D_Y^{\text{sorted}}$ obtained at iterations $i, j$, respectively. Let their respective labeling candidates be $X_{y_i} := \{x \in D_X^{\text{sorted}} \mid cos\_sim(x, y_i) = 1\}$ and $X_{y_j} := \{x \in D_X^{\text{sorted}} \mid cos\_sim(x, y_j) = 1\}$. Then if $cos\_sim(y_i, y_j) \neq 1$ (on different rays) then $X_{y_i} \cap X_{y_j} = \emptyset$. In other words, if $y_i$ and $y_j$ reside on different rays, then their labeling candidates are disjoint (note that there is no risk that a given $y$ will be labeled with an $x$ from a different ray since the combined assumptions that each $x$ can be matched with a unique $y$ on the same ray, and that $|D_Y| = |D_X|$, mean that there is an equal number of $Y$ and $X$ samples on each ray and the maximal cosine similarity of 1 will always be realized, necessarily yielding an $x_y$ from the same ray).

Since the labeling Algorithm operates independently on two different rays, we may assume that our random variables are from a single ray $r$, $X^r$ and $Y^r$. We now prove the claim by induction on the number of iterations of Algorithm 1 when applied to $D_{X^r}^{\text{sorted}}, D_{Y^r}^{\text{sorted}}$. In iteration $n = 1$ we have $y = y_1$ where $y_1$ has the smallest norm in $D_{Y^r}^{\text{sorted}}$. Since all $x \in D_{X^r}^{\text{sorted}}$ satisfy $cosine\_sim(x,y) = 1$ and the candidates are sorted by norm, then $x_y = x_1$ where $x_1$ has the smallest norm in $D_{X^r}^{\text{sorted}}$. Since both are the 0-th empirical quantile in terms of norm, we have that $\tilde{h}_1(||y_1||) = ||x_1||$, i.e.

$$\tilde{h}_1(||y||) = ||x_1||$$

Plugging this result and $x_y = x_1$ into our claim, it remains to prove that:

$$x_1 = ||x_1||\frac{y_1}{||y_1||}$$

but since both $x_1$ and $y_1$ are on the same ray, we have that their unit vectors are equal, that is:

$$\frac{y_1}{||y_1||} = \frac{x_1}{||x_1||}$$

so that:

$$x_1 = ||x_1||\frac{y_1}{||y_1||} = ||x_1||\frac{x_1}{||x_1||} = x_1$$

which is a truth statement so the claim is proved for $n = 1$.

At the $n$-th iteration, we must have that $\tilde{h}_1(||y_k||) = ||x_{y_k}||$ for all $k < n$, since by the induction hypothesis:

$$x_{y_k} = \tilde{h}_1(||y_k||)\frac{x_{y_k}}{||x_{y_k}||}$$

and by applying the norm on both sides:

$$||x_{y_k}|| = \tilde{h}_1(||y_k||)$$

Therefore, $y_n$ is the $n$-th empirical quantile in $D_{Y^r}^{\text{sorted}}$ and $x_y$ is the $n$-th empirical quantile in $D_{X^r}^{\text{sorted}}$ (Algorithm 1 removed all previously matched elements from both lists).

Hence, similar to $n = 1$, it remains to prove:

$$x_{y_n} = ||x_{y_n}||\frac{y_n}{||y_n||}$$

and again, since:

$$\frac{y_n}{||y_n||} = \frac{x_{y_n}}{||x_{y_n}||}$$

we have:

$$x_{y_n} = ||x_{y_n}||\frac{x_{y_n}}{||x_{y_n}||} = x_{y_n}$$

which completes the proof.

**Note**: Although for $d > 1$ this assumption occurs with probability 0 with *random* sampling from $Y$, it may be possible to achieve this by other means. For example, given a sample $x \in D_X$, one could obtain a sample from $Y$ that is on the same ray as $x$ by first sampling from $Y$ restricted to the $x$-axis and then rotating the sample to be in the direction of $x$ (this requires $Y$'s distribution to be rotation invariant, e.g. a normal distribution). We note that if such $D_X, D_Y$ are obtained, then there would be no need for the cosine similarity in Algorithm 1, of course, but a different sampling (generating) procedure may be required. We provide a demonstration of this in Figure 9 along with additional details on the method. We leave further development and analysis of this approach to future research.

## D    Topological properties

One example of a topological property is the connectedness property – being a "single piece", like $\mathbb{R}$, as opposed to "more than one piece", like $(-\infty, -1) \cup (1, \infty)$, is preserved between homeomorphic spaces (so, in particular, these two are not homeomorphic, but $\mathbb{R}$ *is* homeomorphic to each of the two intervals, separately).

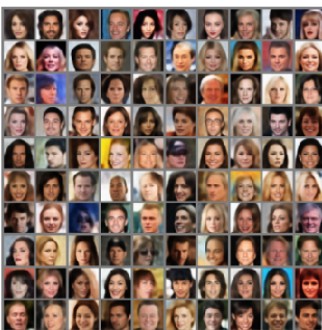 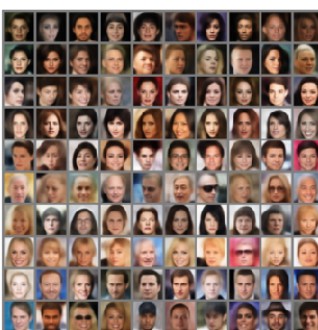

Figure 9: Random real reconstructions and random generated samples for $d = 100$ after using Algorithm 1 with $D_Y$ as mentioned in the accompanying note to Appendix C (left is real). Specifically, $D_Y$ is sampled from a mixture of 20 Gaussians, each centered around a cluster mean (see Figure 7 for details), where each normal sample was rotated to be on the ray of some $x \in D_X$ in its associated cluster (the one which determines the mean of its Gaussian). Algorithm 1 was then applied to these $D_X, D_Y$, which now satisfy the condition that each ray contains the same number of points from $X$ and $Y$. To generate a sample, new $Y$ points were obtained in a similar fashion, i.e. by randomly sampling a new point from the Gaussian mixture and rotating the sample to a random ray in its associated cluster.

Therefore, if we suspect that our data is composed of separate classes like images of both hands and faces, and we would like to train a GTN as a generative model, one option would be to use a clustering approach and to generate labels for each cluster separately. Another would be to split our data into its different components (learning a separate $h$ for each). This relates to the conditions of Theorem 2.1.1 which assume that $X$'s support is an open interval. For example, if we suspect that our data can be separated into two disjoint uniform distributions, then the theorem doesn't apply. However, it *does* apply to each separate uniformly distributed component.

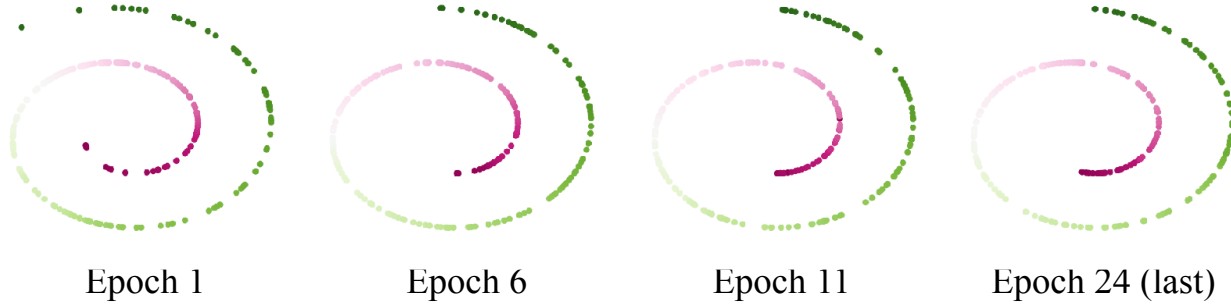

Figure 10: Swiss-roll samples generated by GTN during the training process.

## E   $h$ as defined in Section 2.2.1 is a homeomorphism

$h$ is injective: equating for two points $y_1, y_2$ we obtain:

$$h_1(||y_1||)\frac{y_1}{||y_1||} = h_1(||y_2||)\frac{y_2}{||y_2||}$$

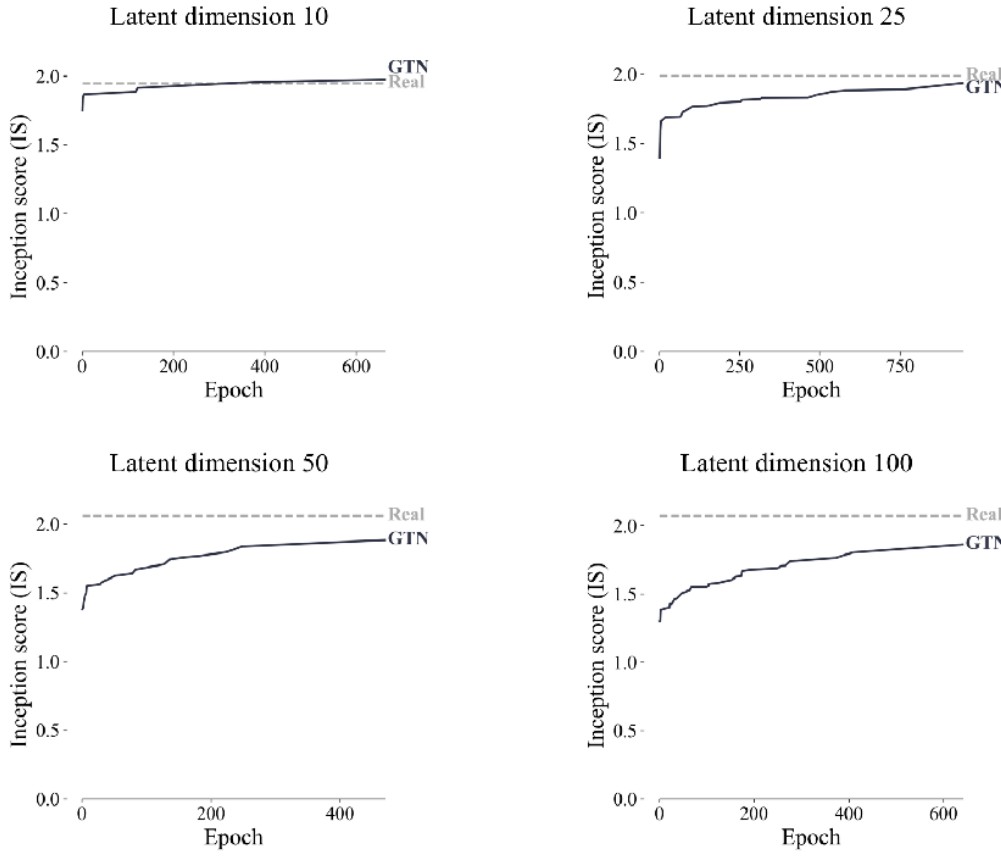

Figure 11: Inception score (IS) vs recon-IS for CelebA plotted by epoch (at epochs of improvement in the IS). At each epoch, the IS was computed for 200 randomly generated images (decoded latent vectors generated by $\hat{h}$ from random normal vectors, formally: *autoencoder.decoder* $\left(\hat{h}(r)\right)$ for $r \sim \mathcal{N}(\mathbf{0}, \mathbf{I})$) (solid black line). recon-IS was computed once for a random set of 200 reconstructions (dashed gray line).

Applying the norm to both sides yields $h_1(||y_1||) = h_1(||y_2||)$. Since $h_1$ is injective, this means that $||y_1|| = ||y_2||$. Plugging this into the equality and cancelling out equal terms ($y_1, y_2 \neq 0$) yields $y_1 = y_2$. The function is also onto: since $h_1$ is onto (it is a homeomorphism) then for any $x \neq 0$ there is a value $v$ in $(0, \infty)$ with $h_1(v) = ||x||$. Since the support of $Y$ is $\mathbb{R}^d$ there is a $y$ that is on the same line as $x$ from the origin and that satisfies $||y|| = v$ so that: $x = h_1(||y||)\frac{y}{||y||}$, meaning that $x$ has a $y \in S_Y$ for which $h(y) = x$.

$h$ is continuous since for $y \neq 0$ it is a composition of continuous functions, and for $y = 0$ we observe that $\lim_{y \to 0} h_1(||y||)\frac{y}{||y||} = 0$ ($\frac{y}{||y||}$ is the unit vector in the direction of $y$ and $h_1(||y||) \to 0$ as $y \to 0$ by definition of the CDF).

The inverse is also continuous since a continuous and bijective function between open subsets of $\mathbb{R}^d$ is continuous (by the invariance of domain theorem).

## F   A line-based point-of-view

Another way of thinking about this is as follows: assume for simplicity that $f_X, f_Y > 0$ (so that $S_X, S_Y = \mathbb{R}^d$ and therefore $l \cap S_X$ and $l \cap S_Y$ are simply $l$). Since $l$ is a 1D manifold, $X$ and $Y$ induce 1D random variables

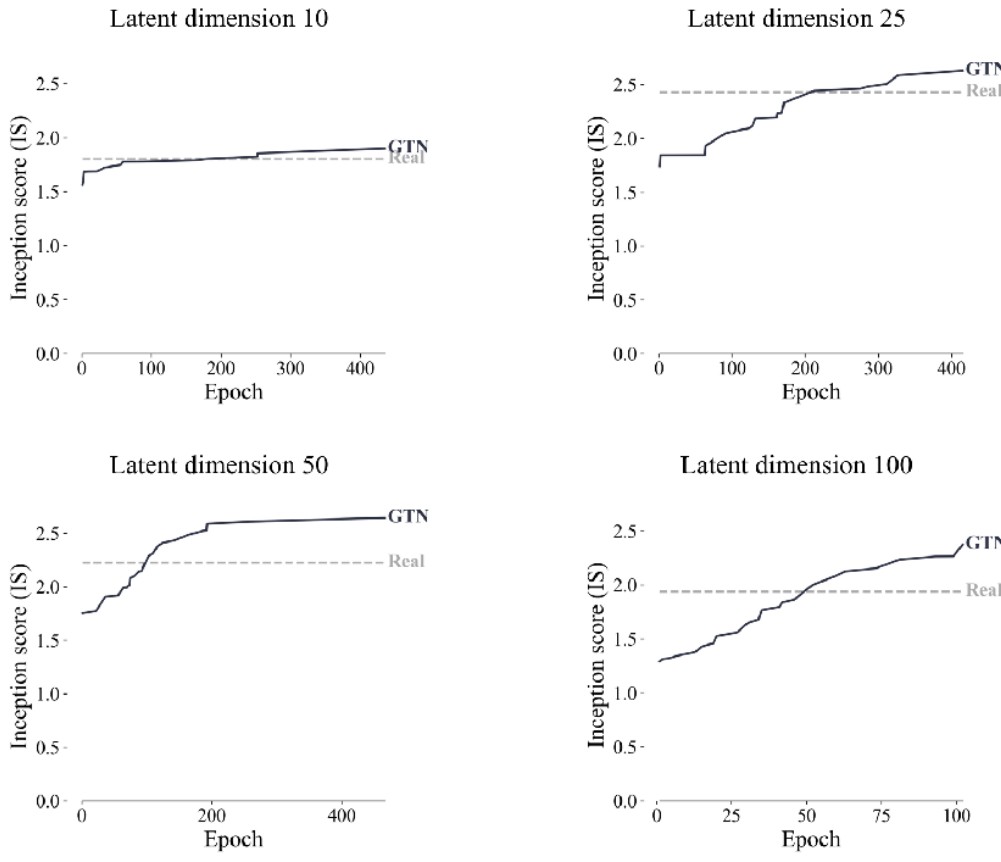

Figure 12: Inception score (IS) vs recon-IS for HaP plotted by epoch (at epochs of improvement in IS). At each epoch, IS was computed for 200 randomly generated images (decoded latent vectors generated by $\hat{h}$ from random normal vectors), formally: $autoencoder.decoder\big(\hat{h}(r)\big)$ for $r \sim \mathcal{N}(\mathbf{0}, \mathbf{I})$) (solid black line). recon-IS was computed once for a random set of 200 reconstructions (dashed gray line).

$X^l$ and $Y^l$ on the line $l$, with pdfs $f_{X^l}, f_{Y^l}$ and CDFs $F_{X^l}, F_{Y^l}$. We can apply Theorem 2.1.1 to these random variables to obtain a homeomorphism $h^l$ for each line.

## G  Architecture and training specifications

The same autoencoder architecture was used for MNIST, CelebA and HaP with only the essential modifications needed to accomodate to the different input channels (1 for MNIST and 3 for CelebA and HaP) and the different latent dimensions. The architecture consisted of a decoder with two 2d convolutional layers followed by ReLU activation: the first convolutional layer had 64 output channels, kernel size 4, stride 2 and padding 1. The second convolutional layer consisted of 128 output channels, kernel size 4, stride 2 and padding 1. The two convolutional layers were followed by a liner layer with the number of output features set to the desired latent dimension $d$. The output activation was tanh. In the decoder we used a mirror architecutre of one linear layer, with the number of input features being $d$, and two 2d transposed convolution layers. The code for all architectures is also available in our repository. To train the autoencoders we used: a batch size of 200 for CelebA and HaP, and 128 for MNIST; learning rate $1e-4$ for CelebA and HaP and $1e^{-3}$ for

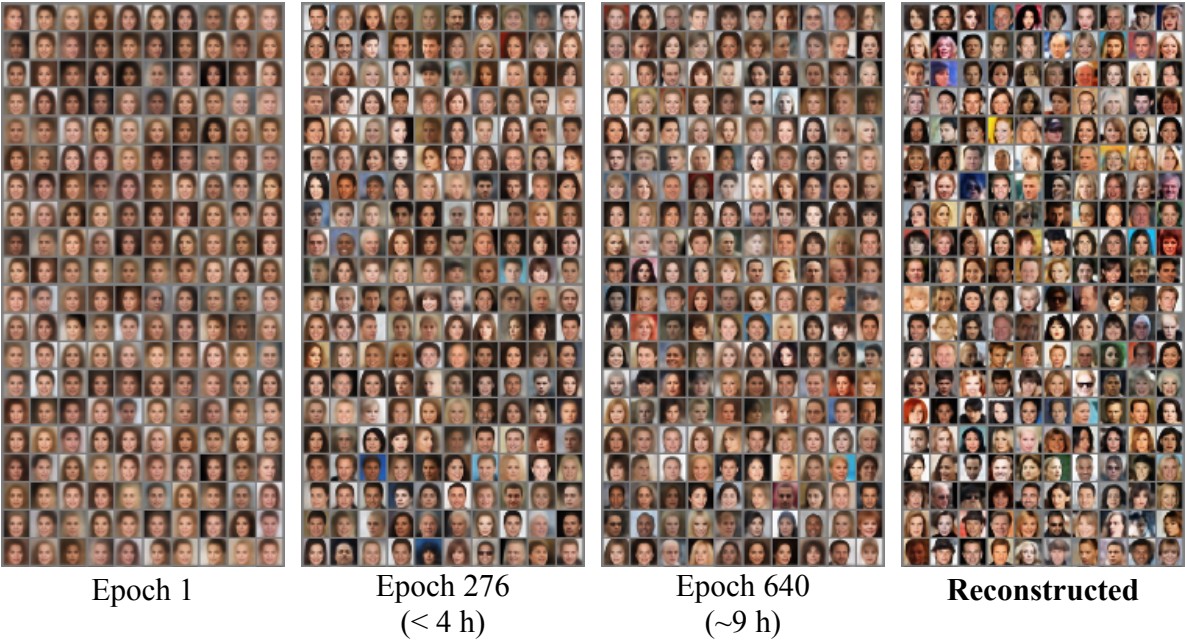

| Epoch 1 | Epoch 276 (< 4 h) | Epoch 640 (~9 h) | **Reconstructed** |

Figure 13: Randomly chosen samples generated by a GTN during the training process on CelebA with latent dimension 100 at several epochs of improvement in the IS score, and randomly chosen real images (bottom-right grid). Specifically, each epoch shows 200 images, each of which is the decoded $\hat{h}(r)$ for some random $r \sim \mathcal{N}(\mathbf{0}, \mathbf{I})$. The bottom right grid shows 200 images each of which is the decoded latent vector of a random real image. Training until epoch 640 took approximately 9 hours on a single T4 GPU, with similar-quality images obtained in less than 4 hours of training at epoch 276.

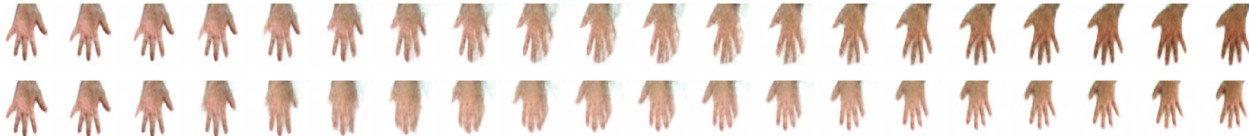

Figure 14: Interpolations between a hand (downward facing) and a palm (upward facing).

MNIST; a weight decay of $1e^{-5}$ was used in all three. We tested the two learning rates $1e^{-3}$ and $1e^{-4}$. No further optimizations were made for the autoencoder hyperparameters.

CelebA images were center-cropped to $148 \times 148$ and resized to $64 \times 64$. HaP images were resized to $64 \times 64$.

We did not use any data augmentations during the training process.

Final architecture specifications for $\hat{h}$ appear in Appendix Table 5. We ran a hyperparameter search for CelebA and HaP. Initially, we ran several settings for CelebA using a small number of epochs (roughly 100-300) for latent dimension 200 prior to consulting the literature on the intrinsic dimension of image data. We tried learning rates of $1e^{-3}, 1e^{-4}, 1e^{-5}, 5e^{-5}$ for various width and depth settings. Specifically, for CelebA we tested widths of $500, 1000, 1200, 1300, 1500$ and depths of $10, 17, 20, 25, 27$. For HaP we kept the depth at 25 after observing this obtained best results for CelebA and tested widths of $2000, 3000$ as well after identifying that HaP benefited from higher width settings.

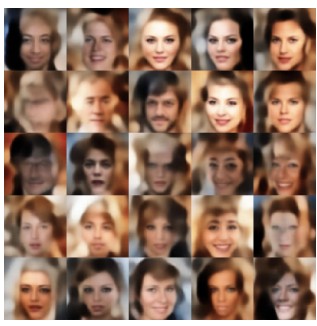 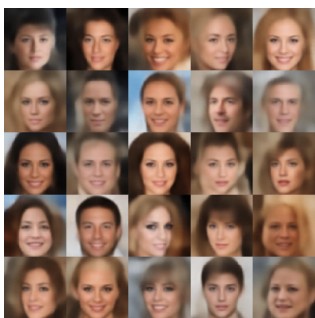

Without GTN
(decoded random normal)

With GTN

Figure 15: Generated images by: (left) decoding random normal vectors with vectors $\mu, \sigma$ estimated from the latent space (each image is the decoded $r$ for some random $r \sim \mathcal{N}(\mu, \sigma I)$); (right) GTN (each image is the decoded $\hat{h}(r)$ for some random $r \sim \mathcal{N}(0, I)$)).

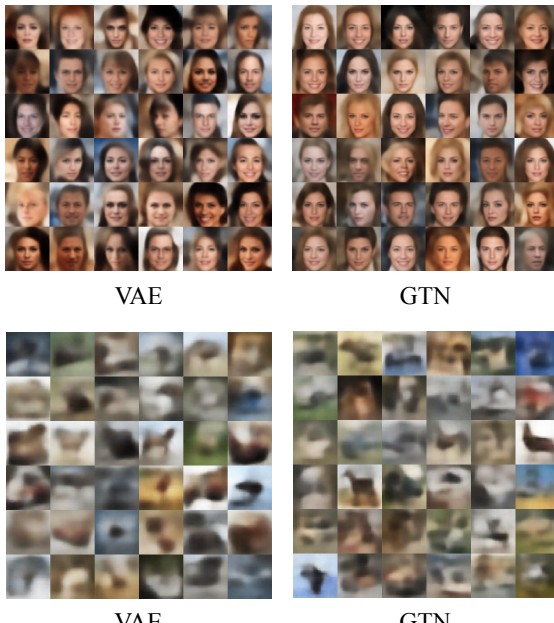

VAE                    GTN

VAE                    GTN

Figure 16: Controlled comparison of GTN with VAE (Kingma et al., 2016) on CelebA and CIFAR-10 – randomly generated images from both.

The architectures were compared based on their IS score at each epoch. The architecture with the highest IS score was kept. These architectures are the ones described in Appendix Table 5.

We used the best settings, shown in Appendix Table 5, regardless of the dimension $d$. We did not perform architecture optimizations per dimension. Such optimizations may provide further improvements in efficiency and/or generative quality.

| | CelebA | | | | HaP | | |
|---|---|---|---|---|---|---|---|
| | IS ↑ | recon-IS ↑ | FID ↓ | | IS ↑ | recon-IS ↑ | FID ↓ |
| $d = 10$ | 1.97 | 1.95 | 119.46 | | 1.90 | 1.80 | 192.98 |
| $d = 25$ | 1.93 | 1.99 | 82.88 | | 2.63 | 2.43 | 123.89 |
| $d = 50$ | 1.88 | 2.06 | 71.61 | | 2.64 | 2.22 | 81.48 |
| $d = 100$ | 1.86 | 2.07 | 66.05 | | 2.37 | 1.94 | 101.85 |

Table 2: Results per dataset and latent dimension using vanilla autoencoders (2 layer encoder). Inception Score (IS) is the best IS attained during training for the given dimension and dataset for a random set of 200 reconstructions. recon-IS is the IS attained by a set of 200 reconstructions. Fréchet Inception Distance (FID) is computed using $50,000$ randomly generated images and $50,000$ random real images using the same model that attained the reported best IS. For the smaller HaP dataset, all training images ($\sim 8,000$) and the same number of randomly generated images were used.

Each experiment was run on 1 NVIDIA T4 GPU with 8 vCPU + 52 GB memory, 500GB SSD. Specifically, we used the "Deep Learning VM" by Google Click to Deploy in the Google Cloud Marketplace (image: pytorch-1-13-cu113-v20230925-debian-10-py37), modified to the aforementioned specifications.

## H   Generating 1D swiss-roll data

The 1D swiss-roll data is generated by sampling $\theta \sim U(1.5\pi, 4.5\pi)$ and computing $f(\theta) = \theta(\cos\theta, \sin\theta)$. GTN is trained to generate $\theta$. This example, besides demonstrating the 1D case, serves to demonstrate how GTN operates on the 'correct' latent representation of the data (in practice, this could be obtained by an appropriate dimensionality reduction method, like an autoencoder).

## I   Significance of $h$ being a homeomorphism for data synthesis

The fact that $h$ is a homeomorphism is significant in the context of generative models for several reasons:

1. **Learnability** – Since $h$ being a homeomorphism implies that both $h$ and $h^{-1}$ are continuous real-valued functions over some subset of $\mathbb{R}$, then by the universal approximation theorem they can be approximated to arbitrary accuracy by a neural network (Hornik et al., 1989). *This is demonstrated in Figure 3 (A).*

2. **Coverage and diversity** – The bijectivity of a homeomorphism means that there is a one-to-one and onto correspondence between the supports $S_X$ and $S_Y$. This means that we can use $Y$ to cover all samples that can be obtained from $X$ and that no two samples in $Y$ will generate the same sample in $X$. *This is demonstrated in Figure 3 (A) and (B).*

3. **Continuous interpolation** – The fact that $h$ is continuous is significant for purposes of interpolation. For example, given a generative model $g : \mathbb{R} \to S_X$, two points $y_1, y_2 \in \mathbb{R}$, and the function: $\phi(\lambda) = \lambda y_1 + (1-\lambda)y_2$ (where $\phi : [0, 1] \to \mathbb{R}$) we would like $(g \circ \phi)(\lambda)$ to be continuous (e.g. for video generation). If $g$ is stochastic, for example, this cannot be guaranteed. However, using $h$ as $g$, we have that $g \circ \phi$ is continuous as a composition of continuous functions. This provides the desired continuous interpolation between points from $X$. *This is demonstrated in Figure 6.*

4. **Guiding topological properties** – There are useful properties that are invariant under homeomorphisms ("topological properties") that can guide us in designing better generative models. For example, one property is that homeomorphic manifolds must have the same dimension. *The use of this property is demonstrated in the upcoming swiss-roll example*, where it will lead us to the conclusion that we are better off generating swiss-roll samples from a 1D standard normal distribution, rather than from a 2D one. Another example for a useful property is in Appendix D.

| Method | FID ↓ |
|---|---|
| 2-Stage VAE (Dai & Wipf, 2019) | 44.4 |
| NVAE (Vahdat & Kautz, 2020) | 14.74 |
| WAE-GAN (Tolstikhin et al., 2017) | 42.0 |
| DiffuseVAE (Pandey et al., 2022) | 3.97 |
| GLF (Xiao et al., 2019) | 53.2 |
| VAE + flow posterior (Xiao et al., 2019) | 67.9 |
| VAE + flow prior (Xiao et al., 2019) | 54.3 |
| VAE + GMM=75 (Pandey et al., 2022) | 72.11 |
| GTN (ours) | 66.05 |

Table 3: GTN results and results reported by related methods on CelebA. GTN results are for $d = 100$ (from Table 2). All other results are as originally reported by their authors, except for NVAE which does not report FID (results are as reported by Pandey et al. (2022)). Note that the methods may differ in their evaluation settings and implementation.

| Method | FID ↓ |
|---|---|
| 2-Stage VAE (Dai & Wipf, 2019) | 72.9 |
| NVAE (Vahdat & Kautz, 2020) | 51.67 |
| DiffuseVAE (Pandey et al., 2022) | 2.62 |
| FM-OT (Lipman et al., 2022) | 6.35 |
| GLF (Xiao et al., 2019) | 88.3 |
| VAE + flow posterior (Xiao et al., 2019) | 143.6 |
| VAE + flow prior (Xiao et al., 2019) | 110.4 |
| VAE + GMM=75 (Pandey et al., 2022) | 137.68 |
| GTN (ours) | 238.62 |

Table 4: GTN results and results reported by related methods on CIFAR-10. GTN results are from Table 1. All other results are as originally reported by their authors, except for NVAE which does not report FID (results are as reported by Pandey et al. (2022)). Note that the methods may differ in their evaluation settings and implementation.

| | No. Hidden Layers | Width | Activation | Batch Norm. | Learning Rate | Weight Decay | Optimizer | Batch Size |
|---|---|---|---|---|---|---|---|---|
| Swiss-Roll | 4 | 6 | LeakyReLU(0.5) | No | $1e^{-3}$ | No | Adam | 250 |
| Uniform | 6 | 6 | LeakyReLU(0.5) | No | $1e^{-3}$ | No | Adam | 250 |
| MNIST | 7 | 50 | LeakyReLU(0.5) | Yes | $1e^{-3}$ | No | Adam | 128 |
| CelebA | 26 | 1,200 | LeakyReLU(0.5) | Yes | $5e^{-5}$ | No | Adam | 200 |
| HaP | 26 | 3,000 | LeakyReLU(0.5) | Yes | $5e^{-5}$ | No | Adam | 200 |
| CIFAR-10 | 26 | 1,500 | LeakyReLU(0.5) | Yes | $5e^{-5}$ | No | Adam | 200 |

Table 5: Architecture specifications for $\hat{h}$ for the different datasets. The architecture was kept the same between the different dimensions where different dimensions were tested (CelebA, HaP).

