# OpenReview forum: "Generative Topological Networks"
_TMLR — Rejected by TMLR_

### Review · Reviewer_gPYY · 2025-02-10

**Summary Of Contributions:**

The paper introduces Generative Topological Networks (GTNs), a novel method for generative modeling. GTNs have a simple formulation and is simple to train, as they are trained in the form of a supervised learning method. The paper demonstrates that GTNs yield reasonably good performances compared with VAE on several datasets when trained using the latent representation of the data. The paper highlights the importance of topological properties for generative modeling techniques.

**Audience:**

Yes

**Broader Impact Concerns:**

Not applicable.

**Claims And Evidence:**

Yes

**Requested Changes:**

It might be worth better highlighting the benefits of the proposed approach compared with the other methods, e.g. VAEs and flows based methods; for instance, does it result in computational speedups? VAEs are shown to yield (much) better performances than GTNs in the paper, while there were no flows results.

**Strengths And Weaknesses:**

Strengths: 1. The paper is generally well-written, and the algorithms are introduced in the order of the complexity of the setting, allowing the readers to understand the ideas better. 2. The method has a simple formulation, while achieving improvements upon VAE.

Weaknesses: 1. Looking at Table 3 and Table 4 it appears GTNs does not yet yield comparable performances compared with SOTA VAEs. While this can be attributed to the lack of tunings, it is nonetheless a weakness. 2. Only VAE-like methods are considered as baselines. Normalizing flows methods, while being referred to multiple times in the paper, do not seem to be compared against. While the paper highlights the architectural constraints of normalizing flows, it is unclear whether GTNs yield superior results over flows. 3. The paper claims that GTNs do not suffer from pitfalls like mode collapse. While it makes some sense by looking at the algorithms, it was not empirically validated (?).

---

> ### Author Response · Authors · 2025-03-10
>
> We thank the reviewer for indeed raising important points and for the kind and useful comments. Regarding weakness 1, we actually suspect that the main cause for the underperformance of GTN is in using simplistic autoencoders (we, in fact, provide both quantitative and qualitative evidence in the paper that the majority of the error stems from reconstruction). In future work we plan to explore stronger representations. Regarding weakness 2, we will try to include more comparisons to normalizing flows. As for 3 – it is not clear to us how to prove the absence of mode collapse, but we can provide a large sample in the github repository that empirically demonstrates this. We agree that the comparisons are far from comprehensive. We did, however, provide a controlled experiment to compare against VAE and provided both qualitative and quantitative conclusions. We will try to expand the comparison to also cover normalizing flows.

---

### Review · Reviewer_Y8eK · 2025-02-27

**Summary Of Contributions:**

This work proposes a new generative model, the Generative Topological Network (GTN). The method is presented in two stages:
1. The 1D case. The idea is to train a neural network $\hat{h}$ to match the quantiles of $X$, the data distribution, and $Y$, a base distribution (e.g. a Gaussian). This is done by grouping the sorted samples $x_1, \ldots, x_n \sim X$ and $y_1, \ldots, y_n \sim Y$ into pairs $(x_i, y_i)$ and training  a neural network $\hat{h}$ to map $y_i$ to $x_i$ via MSE.
2. the nD case (Algorithm 1). Since the 1D approach does not generalize to higher dimensions, the authors need to propose another way of pairing samples. This is done by iterating through $y_1, \ldots, y_n$ and pairing it with the nearest $x$ according to cosine similarity. A neural network $\hat{h}$ is then trained again to map samples $y_i$ to $x_i$. The resulting model is the GTN.

The work also points out that the topology (in the sense of intrinsic dimension) of ground truth distributions is often ignored by generative models, and therefore GTNs are trained in most cases in the latent space of an autoencoder.

The resulting model is evaluated on a number of image datasets.

**Audience:**

No

**Claims And Evidence:**

No

**Requested Changes:**

Work needs a complete overhaul at best, but its fundamental premise is itself questionable. See strengths and weaknesses.

**Strengths And Weaknesses:**

There are too many specific issues with this paper to list exhaustively. In hopes it will help the authors iterate on their research direction, here's an attempt to distill them into coherent themes.

I see three overarching problems with the work:

1. The provided algorithm/model for building GTNs is not well-justified. There is some nice theory the authors present from pages 3-6 for simple cases, but there is then a huge leap to get to the actual implementation. This means the actual model evaluated in the paper is nearly irrelevant to most of the theoretical claims. I also don't think the final model *per se* makes sense. I doubt this can be salvaged (see discussion on Algorithm 1 below).

2. The experiments do little to justify the GTN’s usefulness as a generative model. Baselining is weak and there is no strong reason, beyond some unsupported claims made in the intro and conclusion, to believe it is more useful in any respect than any of the generative models used today.

3. The work also suggests a lack of awareness of the literature and its place within, including some mildly incorrect or outdated claims (e.g., evaluating with inception score, which has long been disavowed by ML researchers [H], or fixating on (1) GAN mode collapse or (2) VAE posterior collapse, which are (1) basically solved and (2) irrelevant to this context, respectively) and missing huge swathes of relevant modern context from the generative modelling scene (e.g. flow matching? manifold learning in the context of generative modelling?).

Below is a nonexhaustive list of specifics.

Presentation and claims:
- The progression of ideas in this work needs to be streamlined. The method is motivated over the course of 3 pages by first presenting (1) the 1-dimensional case and (2) an n-dimensional rotation-invariant case. I would contend that the way these inspire the final method is fairly superficial. For instance, the work writes that "the simpler case above provides guidance on how to train a neural network to match between two distributions on each line" (where "line" here refers to a ray emanating from the origin), but then in the final method (Algorithm 1), no serious attempt is made to match distributions on each ray (see my concerns about Algorithm 1 below). You may want to consider cutting much of this motivation to spend more space discussing and analyzing the final algorithm.
- Some claims feel odd or out-of-place. For example, in section 2, when describing the 1D method, there is a digression about how DGMs in ambient space are poor models for low-intrinsic-dimension data. I agree with this point in general, but I have two issues with the way it is presented here.
	- I am skeptical of the given example, where a diffusion model fails to learn a toy 1D swiss roll example. In my experience, diffusion models can learn toy, low-dimensional manifolds when adequately tuned. There is theoretical work in the literature to back this up [A]. Did you try tuning the hyperparameters of Jimenez (2024)? Training for more epochs or increasing model capacity might help.
	- While you seem to be framing this observation about intrinsic dimensionality mismatch as a novel contribution, it is actually a well-known phenomenon that has been studied for a while. You do not need to spend so much space illustrating this point; you can simply cite e.g. [B] and [C], and move on. You should also tone down your claims around this, e.g., that this work "provides a new perspective on the importance of topological considerations like the intrinsic dimension."
	- It is also claimed that GTNs avoid issues like mode collapse or posterior collapse faced by GANs and VAEs, but the given experiments are not nearly large-scale or thorough enough to support this.

Algorithm 1:
- You seem to be claiming that $\hat{h}$ is a homeomorphism. While I agree this is true in the 1D and rotation-invariant cases, it is not at all clear to me that Algorithm 1 should produce a homeomorphism in the general case. Do you have any further justification for this?
- Algorithm 1, which is meant to match nearby points from $D_X$ and $D_Y$ together, appears to entirely disregard their norms in practice and only account for their cosine similarity. Using the norm to settle ties would never address this problem in a real situation since the probability of ties in cosine similarity is typically 0. The paper obliquely points this out, but does not address it.
- Why match points using cosine similarity and not L2 distance?

Experiments:
- Figure 3 (a) of the swiss roll is produced by using GTNs to learn a uniform distribution in 1D space and then projecting them through the pre-defined swiss roll embedding. This isn't a very informative experiment, as the pre-defined embedding is already doing all the heavy lifting of defining the manifold.
- The GTN's performance on the uniform distribution in Figure 3(b) is not particularly impressive.
- The main baseline is a Gaussian VAE (in the sense that $p_\theta(x | z)$ is Gaussian. Gaussian VAEs aren’t really used in practice as they are known to produce blurry images [B]. If you are looking for an autoencoder-based architecture, why not try something stronger, like a Wasserstein autoencoder [D] for instance?
- Also, crucially, it is well known that generative models trained in the latent space of a auto-encoder-type models (i.e., two-step models [C]) tend to be highly performant, regardless of specific architecture. Yours has an autoencoder component and thus is a two-step model; you should probably use two-step baselines rather than a simple VAE to convincingly demonstrate the efficacy of your method.

Additional citation suggestions
- The method's CDF-based motivation bears some resemblance to neural inverse transform samplers [E].
- Learning direct mappings between distributions X and Y is in a way similar to flow matching [F], and rectified flows in particular [G].

Typo:
- p5: homermorphic

Sources:

[A] Pidstrigach, Jakiw. "Score-based generative models detect manifolds." _Advances in Neural Information Processing Systems_ 35 (2022): 35852-35865.

[B] Dai, Bin, and David Wipf. "Diagnosing and Enhancing VAE Models." _International Conference on Learning Representations_ (2019).

[C] Loaiza-Ganem, Gabriel, et al. "Diagnosing and Fixing Manifold Overfitting in Deep Generative Models." _Transactions on Machine Learning Research_ (2022).

[D] Tolstikhin, Ilya, et al. "Wasserstein Auto-Encoders." _International Conference on Learning Representations_. 2018.

[E] Li, Henry, and Yuval Kluger. "Neural inverse transform sampler." _International Conference on Machine Learning_. PMLR, 2022.

[F] Lipman, Yaron, et al. "Flow Matching for Generative Modeling." _The Eleventh International Conference on Learning Representations_ (2023).

[G] Liu, Xingchao, and Chengyue Gong. "Flow Straight and Fast: Learning to Generate and Transfer Data with Rectified Flow." _The Eleventh International Conference on Learning Representations_ (2023).

[H] Barratt, Shane, and Rishi Sharma. "A note on the inception score." _arXiv preprint arXiv:1801.01973_ (2018).

---

> ### Author Response · Authors · 2025-03-10
>
> We thank the reviewer for their detailed review. Regarding the summary, specifically point 2, we would like to point out that the method does not pair $Y$ points with $X$ points using only proximity in cosine-similarity, but rather using both the norm and cosine-similarity (to understand why this is essential, imagine the 1D case; see also Appendix B). As mentioned below, we will clarify this in the appendix. See also our comments to kFyF.
>
> Regarding Algorithm 1 specifically, we emphasize that points are matched using both cosine similarity and L2 norm and that the algorithm indeed provides the labeling by Eq 4 under the necessary assumptions (and, in fact, generalizes it). As such, the Algorithm was heavily inspired by it. To further clarify this, we will provide: (1) A proof that the algorithm does indeed produce the labeling by Eq 4 (see Appendix B for the general idea); (2) Several illustrations in 2D explaining why both the norm and cosine-similarity are indeed essential and take-effect.
>
> With respect to the literature, of course there are additional relevant methods and results, even some not mentioned by the reviewer, including optimal transport. We have chosen to focus on two highly related methods in the main part of the paper, but will expand on other methods, including those suggested by the reviewer, in the appendix.
>
> We agree that the intrinsic dimension mismatch is indeed not a new concept (in fact, we cite others on this in the discussion). While we aimed to emphasize that topological properties (preserved under homeomorphisms) overall may be of interest (including connectivity, not only ID), we agree that it shouldn’t be a main focus, and will tone down the language. In this context, we also agree that the 2D swiss-roll will indeed be learned by diffusion (we also note this in: “..it is well-known that diffusion can..”), but not necessarily the 1D swiss-roll since diffusion is intrinsically noisy (even the swiss-roll example that appears in the original diffusion paper by Sohl-Dickstein et al. includes clear out-of-distribution samples).
>
> We thank the reviewer for suggesting Wasserstein autoencoders. This may indeed be an improvement. We will test this in a planned line of future work employing improved autoencoders.
>
> Thanks for suggesting the citations – we will add them.

---

> ### Comment · Reviewer_Y8eK · 2025-03-18
> **Thanks for the response, but improvements still needed**
>
> I thank you for your response, but I'm afraid it does not alleviate any concerns of mine.
>
> > we would like to point out that the method does not pair $Y$  points with $X$ points using only proximity in cosine-similarity, but rather using both the norm and cosine-similarity
>
> As I pointed out in the original review, the only way the norm is used in Alg. 1 is to settle ties in cosine similarity. Ties will happen for real data with probability near zero, therefore the norm is effectively not used. That is, unless you sample enough training datapoints relative to your numerical precision, which appears to be the case for your rectangle example.
>
> Hopefully if I am misunderstanding something here, you can point it out with specifics.
>
> >  To further clarify this, we will provide: (1) A proof that the algorithm does indeed produce the labeling by Eq 4 (see Appendix B for the general idea); (2) Several illustrations in 2D explaining why both the norm and cosine-similarity are indeed essential and take-effect.
>
> Have you added any such proof to the manuscript yet? I couldn't find it. I should point out that the appendices have no labels or structure - they would benefit from some proper section, theorem, and proof titles where appropriate.
>
> To be clear, I do not contest that it is possible to learn $h_1$ in Eq. (4), but I do not think in practice that Algorithm 1 will give the appropriate labelling with positive probability to any finite data sample as per my concerns above.

---

> > ### Author Response · Authors · 2025-03-24
> > **Thank you, revised version now uploaded.**
> >
> > We thank the reviewer for their response and for being patient with us while we worked to prepare the revised version. We have now uploaded a new version which includes the following:
> >
> > * Section 2.2.4 (along with Figure 4) which shows why the norm takes effect. There is more to it than to settle ties in cosine similarity. In that same section we also explain how both cosine-similarity and the norm take effect together, and why both are required. We are hopeful that this clarifies the interplay between the two.
> > * We have added the proof for Algorithm 1 in Appendix C (as mentioned in our previous comment, it is in line with the general idea provided in Appendix B, which requires certain assumptions). We understand where your concern stems from, and would like to emphasize that you are correct that a random set of finite points for $d>1$ would mean that Algorithm 1 is an approximation of $h$, and not precisely $h$ (with probability 1), but it would still be consistent with $h$ under the described conditions, which have to do with the data obtained from $X$ and $Y$. In light of this, we add a note on a potential modification to the $Y$ sampling procedure which may be interesting to investigate in future research (see below proof).
> > * We have drastically toned down the emphasis on the intrinsic dimension of the data.
> > * We have expanded the related-work section to include flow-matching, optimal transport, rectified flows and more on the manifold assumption. We thank the reviewer for the many useful references.
> > * Regarding IS – we have added many avenues for potential improvements to GTNs in the discussion, one of which is replacing the IS as the stopping-criteria. We would like to note that we were aware of the paper by Barratt et al., but that IS worked as a better stopping-criteria than the MSE on the validation set (as we mention in the paper). Since we use it as a stopping criteria it makes sense to report it, especially since others in the community still do (e.g. [A], [B]).
> > * We have added titles to each section in the appendix for improved readability.
> >
> >
> > [A] Bar-Tal, Omer, et al. "Lumiere: A space-time diffusion model for video generation." SIGGRAPH Asia 2024 Conference Papers. 2024.
> >
> > [B] Ma, Xinyin, Gongfan Fang, and Xinchao Wang. "Deepcache: Accelerating diffusion models for free." Proceedings of the IEEE/CVF conference on computer vision and pattern recognition. 2024.

---

> > > ### Comment · Reviewer_Y8eK · 2025-03-28
> > >
> > > Thanks you for addressing several of my points, but my core concern about the underlying algorithm remains. The proof provided in App. C relies heavily on unrealistic assumptions, such as the ground truth distribution being rotation invariant and the latent and data points lying on the same rays. The brief discussion below the proof does not satisfactorily address these issues, either.

---

> > > > ### Author Response · Authors · 2025-03-29
> > > >
> > > > Thank you for clarifying and editing your previous comment. It is true that under any non-trivial distributions the probability of getting two points on the same line is zero. It is also true that we are using this assumption in the proof. However:
> > > >
> > > > * When the sample size is large enough we will have points positioned at very small angular distances. This is sufficient, in practice, to generate useful representations, as we are showing.
> > > > * Even when working with small sample sizes, the sorting-by-norm step is essential to the functionality of the algorithm. Following the reviewer’s comments, we have added this in Figure 4, specifically A1 and A2 where the entire dataset consists of precisely two points from $Y$ and two points from $X$. We thank the reviewer for incentivizing us to add this clarification to the manuscript.
> > > > * We agree that for quality performance of our approach it will be desirable to have a large number of samples.
> > > > * An additional approach to force 1D collinearity (source and target points on the same ray) is pointed to in the note that accompanies the proof in Appendix C. We have now, in fact, tested this approach, motivated by the reviewer’s comments, for which we are thankful.  While more investigation is required to understand the theoretical grounds, we did implement this approach to images and now show an example of this with an added reference in the note of Appendix C (see newly uploaded revision -- Figure 9). We do note, in addition, that for this approach to be successful, the $Y$ distribution needs to be a mixture of rotation invariant distributions (e.g. a mixture of spherical Gaussians, as in our provided example). We intend to further investigate this direction in future work.
> > > >
> > > > As to the reviewer’s position with respect to interest to the community. Indeed, as pointed out in the review, there are many approaches to generative AI, including some that may be related to GTN. Therefore, we believe that audiences that investigate such approaches will be interested in the methods we present and demonstrate.

---

> ### Author Response · Authors · 2025-03-23
>
> Moved from "comment" to a "reply" to clarify that it is a response to the reviewer's last comment.

---

### Review · Reviewer_kFyF · 2025-03-01

**Summary Of Contributions:**

In this submission, the authors tackle the task of generative modelling, that is to parameterise and learn a probability distribution given a set of samples assumed to be iid and coming from a distribution of interest. In particular, here they are interested in efficient sampling from the trained generative model, and not in likelihood evaluation.
In one dimension the authors propose to learn the mapping $h = F_X^{-1} \circ F_Y$ to map the distribution of $Y$ to $X$. They then resort to defining a joint $p(Y, X)$ by sorting samples ${y_1, \dots, y_n}$ and ${x_1, \dots, x_n}$, and to minimise the MSE $\mathcal{L}(\theta) = \mathbb{E}[\| X - h_\theta(Y) \|^2]$.
They then extend their method to higher dimension by defining a joint by pairing data and noise samples by cosine similarity. Then by assuming that the distribution is $\mathrm{SO}(n)$ invariant in $\mathbb{R}^n$ they fall back to a one dimensional problem where they only need to model the norm of samples (Eq 4).
They eventually showcase they proposed approach on computer vision tasks.

**Audience:**

Yes

**Claims And Evidence:**

No

**Requested Changes:**

The authors need to prove that, assuming the network can be perfectly fitted to minimise the proposed loss, the induced generative distribution is equal to the data distribution.

It would be worth discussing related methods such as Consistency Models (Song et al. 2023) and Flow Map Matching (Boffi et al., 2024) which are proposing to learn (or distill flow models) into a 1 (or few) steps generative model, by approximating the flow map, as manuscript is proposing to achieve.

## Questions
- Section 2.2.1: Does not work for non rotationally equivariant distribution, how is this different from assuming that the data is actually supported in a one dimensional space?
- Section 2.2.2:
    - In the limit of infinite dimension all cosine dimensions will be 0. This method is applied to a latent space, so the dimension is finite and not too large, but still for e.g. SOTA latent diffusion models the latent dimension is several thousands, so won't the cosine similarity be meaningless even in such dimension?

**Strengths And Weaknesses:**

I am not convinced that the proposed method is sound.
This proposed method is reminiscent of  flow matching (Albergo et al. 2023, Lipman et al. 2022) and in particular mini-batch OT coupling (Tong et al. 2023). Yet, here the network is learnt to denoise in 1 step, which intuitively seems like a really hard task.
Yet, in the one dimensional case there is no proof that the proposed method and associated loss will actually fit the data distribution if minimised to the point of reaching a global minimum. Minimising the MSE leads to learning the conditional expectation $h_{\theta^*} = \mathbb{E}[X|Y=y]$. The pushforward $p_\theta = h_{\theta^*} \circ p_Y$ has density given by the change of variable formula $p_\theta(x) = |\frac{d}{d y}h_{\theta^*}(y)|^{-1} p_Y(y)$ with $x=h_{\theta^*}(y)$ but it is unclear why should this be equal to $p_{X}$. A proof would be welcome.
Second, for the multi-dimensional extension is limited to rotationally invariant distribution since from Eq 4, $h$ is only acting on the norm of $y$ and not its direction. This seems like a very strong assumption as removing any directionality means this method is only learning a distribution over the norm which is effectively assuming that the distribution lives in 1D.
What's more, the method requires pair of tuples $(0, X, Y)$ that are aligned, yet this will happen with probability 0.

---

> ### Author Response · Authors · 2025-03-10
>
> We thank the reviewer for detailed and careful comments. Below see our response to the points in the review.
>
> **Summary:**
>
> We are interested in efficient learning of the representation distribution and not only in efficient sampling.
> Also please note that we demonstrate results on distributions that are not $SO(n)$ invariant, such as the 2d-uniform distribution. It's true that we use invariance for the theoretical proof but the method itself generalizes this. We leave further theoretical analysis to future work.
>
> **Comments and questions:**
>
> Our 1D approach is based on matching quantiles between the two distributions (note that Algorithm 1, when applied in the 1D setting, produces a labeling that is equivalent to quantile matching). If the number of samples is large then this mapping will, indeed, approximate the $X$ distribution. In the general case (for any dimension), as $h$ from Eq 4 is a homeomorphism, it is learnable. Therefore a standard neural network can approximate it to arbitrary precision. The question remains as to whether Algorithm 1 indeed labels according to Eq 4 (under its associated assumptions of course). If it does, then since the labeling function is learnable, $\hat{h}$ can approximate it. Although we provide several explanations throughout the paper that justify why the labeling algorithm would accurately approximate the $X$ distribution given a sufficiently large sample (e.g. Fig 2 right; Appx B), we agree that there is room for a unified and formal proof of this. We will provide this proof in the appendix. We will also supplement the proof with several illustrations in 2D explaining why both the norm and cosine-similarity are indeed essential for accurate labeling and that both take-effect.
>
> As can be seen from Eq 4 our mapping actually retains directionality (notice that $h_1(||y||)$ is used to scale $\frac{y}{||y||}$, and therefore $h$ retains the direction of $y$). It is true that our proof uses $SO(n)$ invariance, and this may be a confusion point. We will try to clarify this in the text.
>
> 2.2.1 – see our earlier comment on the 2d-uniform distribution.
>
> 2.2.2 is a good point. Our work currently operates under the manifold assumption of lower dimensional intrinsic dimension. We demonstrate our approach for latent dimensions in the 100s. We leave higher dimensionality to future work.

---

### Author Response · Authors · 2025-03-11

We thank all the reviewers for their helpful comments. We have identified several common themes which we will address in the revised version. Specifically, we will:

(1) Provide a unified proof that shows why Algorithm 1 provides the labeling in Eq 4, and why $h$ in Eq 4 is learnable. While we have briefly alluded to the former in Appx B, and to the latter in the start of the main body, we can see it would be helpful to unify and formalize these in a single place.

(2) Provide demonstrations of how Algorithm 1 operates in both 1D and 2D, specifically emphasizing how *both* cosine-similarity and the norm act together, why the combination is necessary and how this enforces and generalizes Eq 4.

(3) Expand on related work in the appendix, and on where GTN is currently positioned within the field.

(4) Provide a large set of random samples generated by GTN in an online repository to provide further evidence that supports the claim that GTNs do not suffer from mode collapse.

---

### Author Response · Authors · 2025-03-23
**Revised version uploaded -- summary of changes**

We have now uploaded the revised version. This version includes the items we mentioned both in our previous general comment and in our individual replies to all three reviewers. We would like to thank all the reviewers once again for their very helpful comments. We briefly outline the main changes included in the revision:

* We have added Section 2.2.4 (along with Figure 4) which shows why both the norm and cosine-similarity are required and take effect. We are hopeful that this clarifies the interplay between the two.
* We have added the proof for Algorithm 1 in Appendix C (as mentioned in previous comments, this proof is in line with the explanation in Appendix B which requires certain assumptions). We also highlight a potential line of future research for satisfying the required assumptions, involving a modification to the sampling procedure from $Y$  (appears below the proof).
* We have drastically toned down the emphasis on the intrinsic dimension of the data.
* We have expanded the related-work section to include flow-matching, optimal transport, rectified flows and more on the manifold assumption. We thank reviewer Y8eK for the many useful references.
* We provide many avenues for future research, including potential improvements to GTNs, in the discussion.
* We have added titles to each section in the appendix for improved readability.

As mentioned, upon release of the publication, a large set of random samples will be released, demonstrating empirically that GTNs do not suffer from mode-collapse. We will also share the code and weights for the community to reproduce.

---

### Decision · Action_Editor_5MNq · 2025-03-31

**Recommendation:** Reject

**Comment:**

This paper proposes a type of generative model which generalizes the following idea for 1-d distributions: sort the observed samples, sample an equal number of samples from a Gaussian, sort them as well, and then learn a mapping from the sorted Gaussian samples to the sorted data samples. Sampling can then be done by sampling Gaussian noise and mapping it through the learned function. All reviewers agree that the generalization of this idea to higher dimensions proposed by the authors is not of interest. In particular, the authors assume extremely strong assumptions to prove the correctness of their method. Reviewers pointed out these assumptions are highly unrealistic and also highlighted the experimental results being weak. I agree with the reviewers and thus recommend rejection.

**Audience:**

The topic of the submission is of interest to TMLR's audience.

**Claims And Evidence:**

No. Reviewers unanimously agree that the theoretical justification of the method does not apply to any setting of practical interest.